# Review on Deep Neural Networks Applied to Low-Frequency NILM

Patrick Huber *, Alberto Calatroni, Andreas Rumsch and Andrew Paice

iHomeLab, Engineering and Architecture, Lucerne University of Applied Sciences and Arts, 6048 Horw, Switzerland; alberto.calatroni@hslu.ch (A.C.); andreas.rumsch@hslu.ch (A.R.); andrew.paice@hslu.ch (A.P.)
* Correspondence: patrick.huber@hslu.ch

**Abstract:** This paper reviews non-intrusive load monitoring (NILM) approaches that employ deep neural networks to disaggregate appliances from low frequency data, i.e., data with sampling rates lower than the AC base frequency. The overall purpose of this review is, firstly, to gain an overview on the state of the research up to November 2020, and secondly, to identify worthwhile open research topics. Accordingly, we first review the many degrees of freedom of these approaches, what has already been done in the literature, and compile the main characteristics of the reviewed publications in an extensive overview table. The second part of the paper discusses selected aspects of the literature and corresponding research gaps. In particular, we do a performance comparison with respect to reported mean absolute error (MAE) and $F_1$-scores and observe different recurring elements in the best performing approaches, namely data sampling intervals below 10 s, a large field of view, the usage of generative adversarial network (GAN) losses, multi-task learning, and post-processing. Subsequently, multiple input features, multi-task learning, and related research gaps are discussed, the need for comparative studies is highlighted, and finally, missing elements for a successful deployment of NILM approaches based on deep neural networks are pointed out. We conclude the review with an outlook on possible future scenarios.

**Keywords:** non-intrusive load monitoring; load disaggregation; NILM; review; deep learning; deep neural networks; machine learning

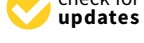



## 1. Introduction

Non-Intrusive Load Monitoring (NILM)—equally referred to as load disaggregation—aims to identify the individual power consumption or on/off state of electrical loads by relying exclusively on the measurement of the aggregated consumption of these loads. The term was coined by Hart in their seminal works [1,2], that initiated the NILM research field. As the term non-intrusive suggests, NILM is motivated by applications where metering of single appliances is necessary, but not feasible with conventional measurement devices, e.g., because of cost considerations or obtrusiveness. Potential NILM applications include, for example, user feedback for energy saving purposes, overseeing activities of daily living for the health assessment of elderly people, or demand management (see, e.g., [3–6]).

Before proceeding, it is essential to clarify some nomenclature typical of the NILM jargon and how this relates to the data used. Electric meters internally sample voltage and current signals at frequencies significantly greater than the base frequency of alternate current (AC). Meters can either output these raw data directly or averaged values such as, e.g., root mean square (RMS) voltage, current, power, or total harmonic distortion (THD) are calculated and output at lower frequencies. In the NILM literature, the following terms are used (We are aware that the given definition is slightly diverging from the conventionally used threshold of 1 Hz (e.g., [3]). However, we feel 1 Hz is somewhat arbitrary, whereas the provided definition seems to be a natural splitting point.):

- Low frequency approaches are those which use data (i.e., features) produced at rates lower than the AC current base frequency;
- High frequency approaches are those which use raw data, sampled at rates higher than the AC current base frequency.

The advantage of using high frequency data should be quite obvious, since these preserve the entire signals and therefore allow us to extract the maximum information content. This is confirmed by various works, e.g., ref. [7] showed that, using raw data sampled at 1 MHz, it is even possible to distinguish between two appliances of the same type. Nevertheless, the cost of gathering high frequency data constitutes a major obstacle: Dedicated infrastructure is needed, and this is expensive both in terms of the hardware itself and the extra installation effort. On the other hand, the loss in information intrinsic in low frequency features is offset by the tremendous ease with which those data can be collected. Indeed, around the year 2010, the European Union and the US started to mandate and actually roll-out smart meters [8]. The advanced versions of these meters can export low frequency data to the outside (e.g., the meter E450 from Landis+Gyr has this capability). The pervasive roll-out of smart meters will therefore unlock all those applications which can benefit from NILM and which can be tackled using low frequency approaches.

Since its inception, the NILM research field has become quite diverse. The classic approach to low frequency NILM—as proposed in [1,2]—is event based. Simply stated, this means that the aggregate power series is first analyzed to find raises or drops that indicate device switching events, and these events are subsequently assigned to certain appliances. Later, one major avenue of NILM research used different variants of Hidden Markov Models, e.g., [9–12]. Over the years, the set of methods that have been applied to NILM has become extremely rich. A comprehensive overview is given in [13]. With the recent enormous success that deep learning has found in the vision and natural language processing domains, it was only a matter of time until deep neural networks (DNN) were also applied for the first time to NILM; this started in 2015 [14,15]. Since then, the number of DNN approaches to solve the NILM problem increased rapidly, as can be seen in Table 1.

**Table 1.** Number of DNN-NILM publications based on low frequency data per year. Numbers are compiled based on Table 2. That means the number for the year 2020 corresponds to the publications until end of November, see Section 1.3.

| Year | 2015 | 2016 | 2017 | 2018 | 2019 | 2020 |
|---|---|---|---|---|---|---|
| **Count** | 2 | 6 | 4 | 21 | 36 | 30 |

Over the years, different publications have surveyed the NILM literature from various angles: In [3], the authors focused on NILM feature types for low and high frequency data and touched algorithms and evaluation metrics. The authors of [16] proposed a taxonomy of appliance features and compared the reported performance of six classes of supervised and unsupervised NILM approaches. The authors of [17] surveyed unsupervised NILM algorithms and discussed the reported performance of eleven approaches. A very comprehensive review on inductive algorithms, employed feature sets, and the state of multi-label NILM classification approaches has been compiled by [13]. More recently, the authors of [18] discussed available public NILM datasets, and employed NILM performance metrics, tools, and frameworks, and corresponding limitations and challenges. Ref. [19] touched on dataset complexity and compared the reported performance of eight NILM approaches under this viewpoint. Three of these approaches employed DNN approaches on low-frequency data. Finally, the authors of [20] summarized advancements on HMM and DNN approaches. With respect to DNN approaches, they summarized the work performed in [21,22]. While the previously mentioned survey papers compared performance as it was reported by the original authors, very recently, two works compared classical and DNN approaches under identical conditions [23,24]. The authors of [23] presented an extension to the NILM tool kit (NILMTK)

library [25,26], with the exact purpose of simplifying direct comparisons, and [24] used this new functionality for an extensive comparison of eight available algorithms.

### 1.1. Contributions

Some of the mentioned review papers include DNN-based NILM approaches. However, what is missing so far is a comprehensive and structured overview on the ideas and findings in the field of DNNs applied to low frequency NILM and, based on that, a discussion on the usefulness of DNN architectural elements, input features or multi-task learning as well as research gaps particularly for applied deep learning-based NILM. In this paper, we intend to fill this gap. Therefore, the main contributions of this work are:

- A comprehensive review of NILM approaches based on low frequency data that employ DNNs, see Section 3. In particular, within Section 3, we discuss various options available for these approaches and provide a structured overview of the main characteristics of all reviewed approaches in Table 2.
- A discussion of selected aspects and corresponding research gaps in Section 4. In particular:
  - We compare the performance of approaches and extract common features of best performing approaches in Section 4.1;
  - We discuss the possible role of multiple input features and multi-task learning on NILM performance in Sections 4.2 and 4.3, respectively;
  - We illustrate the importance of parameter studies in Section 4.4, and
  - We outline major research gaps concerning the application of deep learning for NILM in Section 4.5.

Thereby, we hope that the interested reader will quickly identify the relevant literature for their own research and that our contributions will inspire new research activities, and thus ultimately advance the entire research field.

### 1.2. Scope

The scope of this review are NILM approaches based on DNNs using low frequency data. In the remainder of this text, we use the term DNN-NILM to designate the corresponding approaches. The choice to focus on low frequency data in our review is motivated by our strong belief that many applications could benefit from NILM, coupled with our observation that low frequency data will most likely be the only one available at scale in the near future. In our vision, all households equipped with smart meters will soon be able to become fully energy aware, informing their inhabitants of which appliances are being used, how they are being used, and even whether they are behaving abnormally or about to fail. This latter point is known as predictive maintenance and is currently applied in industrial settings, but being able to detect billions of appliances which consume an abnormal amount of power would have a beneficial impact of our society and its carbon footprint. With our review, we therefore try to make a contribution to push forward the development and understanding of low frequency NILM.

The focus on DNNs is motivated firstly by their proven success in other domains, and secondly by their good performance in the NILM domain: Recently, traditional and DNN-NILM approaches have been compared under identical conditions in two works [23,24]. The authors found that each of the compared DNN approaches—with few exceptions—performed better than each of the classical approaches. In particular for multi-state appliances, the performance gap was found to be "rather discernible" [24]. Publications that use shallow neural networks with only a single hidden layer such as, e.g., [27–29], are not included in our review. We restricted ourselves to approaches that train neural networks with back-propagation, excluding alternative approaches such as, e.g., [30,31]. Since the scope involves DNNs and NILM, we assume that the reader is familiar with the general concepts of the two fields, and we will merely introduce the basic NILM problem formulation in Section 2.1. With respect to DNNs and deep learning, we will refer the reader to comprehensive books on the topic in Section 2.2.

**Table 2.** Reviewed references. Publications are sorted by year. Except for the starred publications, the sorting within a year is arbitrary. The table is available in Excel format on our GitHub account, see 'Supplementary Materials' for the link. Explanations with respect to specific columns follow. **Best**: Best performing, according to Section 4.1. **Datasets**: See Table 3 for details. **Setting**: $R \rightarrow$ residential, $I \rightarrow$ industrial, $C \rightarrow$ commercial. The columns FGE to WDR indicate if the specific appliance has been disaggregated in the reference. **FGE**: fridge, **DWE**: dishwasher, **MWV**: microwave, **WME**: washing machine, **KET**: kettle, **SOC**: stove/oven/cooker, **TDR**: tumble dryer, **HPE**: heat pump, **WDR**: washer-dryer. **Further Appliances**: Additional appliances not listed in the previous columns. **E.Sce.**: Evaluation Scenarios; $sn \rightarrow$ only seen scenario evaluated, $usn \rightarrow$ additionally unseen scenario evaluated, $ctl \rightarrow$ also cross-domain transfer learning evaluated. **Aug.**: Data Augmentation; $dn \rightarrow$ use synthetic training data, $yes \rightarrow$ data augmentation employed. **Input**; $P \rightarrow$ active power, $Q \rightarrow$ reactive power, $I \rightarrow$ current, $S \rightarrow$ apparent power, $P_{2D} \rightarrow$ active power window transformed into 2D representation, $P\text{-}S \rightarrow$ difference between active and apparent power, $\Delta P \rightarrow$ first-order difference of the active power signal, $PF \rightarrow$ power factor, $TofD \rightarrow$ time of day, $WE \rightarrow$ week or weekend day, $DofW \rightarrow$ day of week, $MofY \rightarrow$ month of year, $T_{out} \rightarrow$ outdoor temperature, $P_{var} \rightarrow$ variant power signature [32,33], $na \rightarrow$ see Section 3.3.1. **DNN Elements**: See Section 3.4.1 for the meaning of the various employed abbreviations. Comma separated descriptions refer to different trained models. **Output**: Comma separated descriptions refer to different trained models. Elements connected with an & indicate that a DNN has several outputs of a different type. Similarly, the subscript $m$ means that the DNN provides the identical output for *multiple* different appliances. $on/off \rightarrow$ on/off status of appliance, $P_{class} \rightarrow$ class of active power. Please refer to Section 3.5 for details concerning $P_{app}$, $P_{total}$, $P_{rest}$, *location*, and *stateChange*. **Code**: (electronic version) link to code repository as indicated in the reference or found through a very shallow google search.

| Ref. | Best | Year | Dataset(s) | Setting | DWE | FGE | MWV | WME | KET | SOC | TDR | HVAC | WDR | HPE | Light | Further Appliances | E.Sce. | Aug. | Input | DNN Elements | Output | Code |
|---|---|---|---|---|---|---|---|---|---|---|---|---|---|---|---|---|---|---|---|---|---|---|
| [34] | ★ | 2020 | UK-DALE, REFIT | R | X | X | X | X | X | | | | | | | | usn | no | P | CNN-dAE-GAN | P | https://github.com/DLZRMR/seq2subseq |
| [35] | ★ | 2020 | UK-DALE, REDD | R | X | X | X | X | X | | | | | | | | usn | yes | P | CNN-att-GAN | on/off & P | - |
| [36] | ★ | 2020 | UK-DALE, ECO | R | X | X | X | X | X | X | | | | | | TV | usn | no | P, Q, S, I, PF | CNN-biLSTM | P | - |
| [37] | ★ | 2020 | UK-DALE | R | X | X | | X | | | | | | | | | usn | no | P | CNN-dAE | on/off$_m$ | https://github.com/lmssdd/TPNILM |
| [38] | | 2020 | UK-DALE, REFIT, SynD | R | X | X | X | X | | | | | | | | | sn | no | $P_{2D}$ | CNN-dAE | P | https://github.com/BHafsa/image-nilm |
| [39] | | 2020 | UK-DALE, REDD, REFIT | R | X | | X | X | X | | | | | | | | ctl | no | P | CNN-GAN, CNN-dAE-GAN | P | - |
| [40] | | 2020 | REDD, Enertalk | R | | X | X | X | | | | | | | | TV, rice cooker | sn | no | P | CNN-biLSTM | P | - |
| [41] | | 2020 | REDD | R | | X | X | | | | | | | | | | sn | no | P | CNN-dAE | P | - |
| [42] | | 2020 | REDD, AMPds, REFIT | R | X | | | X | | | X | X | X | X | | | ctl | no | P | biLSTM | P | - |
| [43] | | 2020 | AMPds, REFIT | R | X | | | X | | | X | X | X | X | | | sn | no | P | CNN-dAE-GAN | P | - |
| [44] | | 2020 | proprietary | R | | | | | | | | | | | | water heater | usn | yes | P | CNN-LSTM | P | - |
| [6] | | 2020 | REFIT | R | X | | | X | X | | | | | | | toaster | sn | yes | P | CNN-biGRU | on/off & P | - |
| [45] | | 2020 | proprietary | I | | | | | | | | | | | | | sn | no | p | CNN-biGRU, LSTM | on/off | - |
| [46] | | 2020 | UK-DALE | R | X | X | X | X | X | | | | | | | | sn | no | P | CNN-dAE | on/off$_m$ & P$_m$ | https://github.com/sambaiga/UNETNiLM |

**Table 2.** *Cont.*

| Ref. | Best | Year | Dataset(s) | Setting | DWE | FGE | MWV | WME | KET | SOC | TDR | HVAC | WDR | HPE | Light | Further Appliances | E.Sce. | Aug. | Input | DNN Elements | Output | Code |
|---|---|---|---|---|---|---|---|---|---|---|---|---|---|---|---|---|---|---|---|---|---|---|
| [47] | | 2020 | UK-DALE, REFIT, HES | R | X | | | X | | X | X | | | | | | usn | no | P | CNN-s2p | P | - |
| [48] | | 2020 | REFIT | R | X | | | | X | | | | | | | | usn | no | P | - | P | https://github.com/JackBarber98/pruned-nilm |
| [49] | | 2020 | UK-DALE, REDD | R | X | X | | X | | | | | | | | | usn | no | P | CNN-s2p | P, $P_m$ | https://github.com/EdgeNILM/EdgeNILM |
| [50] | | 2020 | REFIT | R | X | X | | X | X | | | | | | | toaster | usn | no | P | - | P | - |
| [51] | | 2020 | UK-DALE, REDD, DRED | R | X | X | X | | | | | | X | | X | computer | sn | no | P | - | P | - |
| [52] | | 2020 | DRED | R | | X | X | X | | X | | | | | | | sn | no | P | - | P | - |
| [53] | | 2020 | REFIT | R | | | | X | | | | | | | | | usn | no | P | CNN-dAE | P | - |
| [54] | | 2020 | REDD | R | | X | | | | | | | | | | | usn | no | P | LSTM | P | - |
| [55] | | 2020 | UK-DALE, REDD | R | X | X | X | X | X | | | | | | | | usn | no | P | att | P | https://github.com/Yueeeeeee/BERT4NILM |
| [56] | | 2020 | REDD | R | X | X | X | | | X | | | X | | X | bathroom, heater, kitchen outlet | sn | no | P | CNN | on/off$_m$ | - |
| [57] | | 2020 | REDD, dataport | R | X | X | X | X | | | | X | | | X | | sn | no | P, ΔP | CNN-LSTM | $P_m$ | - |
| [20] | | 2020 | summary of [21,22] | | | | | | | | | | | | | | | | | | | |
| [58] | | 2020 | UK-DALE | R | X | | | X | | | | | | | | | usn | yes | P | CNN-s2p | P | - |
| [59] | | 2020 | REFIT | R | X | X | X | X | X | | | | | | | TV | sn | no | P | CNN-GRU | $P_{app}$ & $P_{total}$ & $P_{rest}$ | - |
| [60] | | 2020 | AMPds | R | X | X | | | | | | | | | X | | sn | dn | P, I | LSTM | on/off | - |
| [61] | | 2020 | proprietary (dc) | R | | | | | | | | | | | | dc appliances | sn | no | I | FF, LSTM | $P_{class}$ | - |
| [62] | ⋆ | 2019 | REFIT | R | X | | | X | X | X | | | | | | | usn | no | P | CNN-wn | P, on/off | https://github.com/jiejiang-jojo/fast-seq2point |
| [63] | ⋆ | 2019 | UK-DALE, REDD, REFIT | R | X | X | X | X | | | | | | | | | ctl | no | P | biGRU, CNN-s2p | on/off & P | - |
| [64] | ⋆ | 2019 | REDD | R | X | X | X | | | | | | | | | | usn | no | P | CNN, CNN-LSTM, LSTM | P | - |
| [65] | | 2019 | UK-DALE | R | X | X | X | X | | | | | | | | | usn | no | S | CNN-s2sub | P | - |
| [66] | | 2019 | proprietary | R | | X | | X | | | | | | | | bottle warmer | sn | yes | P | AE/Kmeans-dAE | P, $P_m$ | - |
| [67] | | 2019 | REDD | R | X | X | X | | | X | | X | | | | | sn | no | P | CNN-s2p | on/off | - |
| [68] | | 2019 | AMPds | R | X | | | X | | X | X | X | | X | | | sn | no | P, MofY, DofW, TofD | LSTM-FF | $P_{class}$ | - |

**Table 2.** *Cont.*

| Ref. | Best | Year | Dataset(s) | Setting | DWE | FGE | MWV | WME | KET | SOC | TDR | HVAC | WDR | HPE | Light | Further Appliances | E.Sce. | Aug. | Input | DNN Elements | Output | Code |
|------|------|------|-----------|---------|-----|-----|-----|-----|-----|-----|-----|------|-----|-----|-------|--------------------|--------|------|-------|--------------|--------|------|
| [69] | | 2019 | UK-DALE | R | X | | X | X | X | X | | | X | | | toaster | sn | yes | P | CNN-s2p | P | - |
| [70] | | 2019 | proprietary | R | | | X | | X | X | X | | | | | coffee machine, hair dryer, rice cooker, toaster, blender, iron, disposer | - | no | P | CNN-dAE | *location* | https://people.csail.mit.edu/cyhsu/sapple/ |
| [71] | | 2019 | UK-DALE, REDD | R | | X | | | | | | | | | | | ctl | no | $P_{2D}$ | CNN | on/off | https://github.com/LampriniKyrk/Imaging-NILM-time-series |
| [72] | | 2019 | UK-DALE, AMPds | R | X | X | X | X | X | X | X | | X | | | | usn | yes | P | CNN-att-biLSTM | P, on/off | - |
| [73] | | 2019 | proprietary | R | | | X | | X | | | X | | | X | computer, fan, hair dryer, printer, TV, water dispenser | ? | ? | I | CNN | on/off$_m$ | - |
| [74] | | 2019 | ECO | R | | X | | X | | | | | | | | computer, freezer | sn | no | P, P-S, TofD, WE | FF | P | - |
| [75] | | 2019 | Enertalk | R | | | | X | | | | | | | | rice cooker, TV | usn | no | P, Q | CNN-s2p | P, on/off | https://github.com/ch-shin [1] |
| [76] | | 2019 | dataport | R | X | X | X | X | | | | X | | | | | sn | no | P | CNN-GRU | P | - |
| [77] | | 2019 | PLAID | R | | X | X | X | | | | X | | | X | computer, fan, hair dryer, heater, vacuum cleaner | usn | yes | na | CNN, FF | on/off$_m$ | - |
| [78] | | 2019 | REDD | R | X | X | X | | | X | | | X | | X | bathroom, kitchen outlet | usn | no | P | RNN | P, on/off | - |
| [23] | | 2019 | dataport | R | | | | X | | | | | | | | air, furnace | usn | no | P | - | P | https://github.com/nilmtk/nilmtk/ https://github.com/nilmtk/nilmtk-contrib |
| [79] | | 2019 | AMPds | R | X | X | | X | | X | X | | | | | | sn | no | P | LSTM-FF, GRU-FF | P | - |
| [80] | | 2019 | AMPds | R | | X | | | | X | X | X | X | | | furnace, TV & entertainment | sn | no | na | CNN | P | - |
| [81] | | 2019 | UK-DALE, REDD | R | X | X | X | X | X | | | | | | | | sn | no | P | CNN, biLSTM | on/off$_m$ | - |

**Table 2.** *Cont.*

| Ref. | Best | Year | Dataset(s) | Setting | DWE | FGE | MWV | WME | KET | SOC | TDR | HVAC | WDR | HPE | Light | Further Appliances | E.Sce. | Aug. | Input | DNN Elements | Output | Code |
|---|---|---|---|---|---|---|---|---|---|---|---|---|---|---|---|---|---|---|---|---|---|---|
| [82] | | 2019 | proprietary | R | X | X | | | | X | | | | | | furnace | sn | no | na | FF | on/off$_m$ | - |
| [83] | | 2019 | ECO | R | X | X | | | X | | | | | | | freezer, home theater | sn | no | P | RNN | on/off$_m$ | - |
| [84] | | 2019 | UK-DALE, dataport | R | X | X | X | X | X | | | X | | | | | sn | no | P | CNN-s2s | P | - |
| [85] | | 2019 | UK-DALE, dataport | R | X | X | X | X | X | | | X | | | | | sn | no | P | CNN | P, on/off | - |
| [86] | | 2019 | UK-DALE | R | X | X | X | X | X | | | | | | | | sn | no | P | CNN-LSTM | P$_m$ | - |
| [87] | | 2019 | proprietary | R | X | X | | X | X | | | | | | | coffee filter, coffee machine, TV | sn | yes | P | RNN, CNN-LSTM, CNN-dAE | P | - |
| [88] | | 2019 | AMPds | R | X | X | | X | | | X | | | X | | | sn | dn | P | LSTM-dAE | P, on/off | - |
| [89] | | 2019 | AMPds | R | X | | | | | X | X | | X | | | | sn | no | P, Q, S, I | CNN-s2s | P | - |
| [90] | | 2019 | AMPds | R | X | | | | | X | X | | X | | | | sn | no | P | biLSTM | P | - |
| [91] | | 2019 | UK-DALE, REDD, REFIT | R | X | X | X | X | X | | | | | | | | ctl | no | P | CNN-s2p | P | https://github.com/MingjunZhong/transferNILM |
| [92] | | 2019 | dataport | R | X | X | X | | | | X | X | | | | | sn | no | P$_{2D}$ | RNN, CNN-dAE | P | https://github.com/yilingjia/TreeCNN-for-Energy-Breakdown |
| [93] | | 2019 | UK-DALE, REDD | R | X | X | X | X | X | | | | X | | | | sn | no | P | - | P | https://gitlab.com/a3labShares/A3NeuralNILM |
| [94] | | 2019 | AMPds2 | R | X | | | | | X | X | | X | | | | sn | no | P, Q, S, I | CNN-wn | P$_m$ | https://github.com/picagrad/WaveNILM |
| [95] | | 2019 | REDD | R | X | X | X | | | | | | X | | | kitchen outlet | sn | no | ΔP | FF, biGRU | stateChange | - |
| [96] | | 2019 | REDD, dataport | R | X | X | X | X | | | X | | | | | air, furnace, kitchen outlet | usn | no | P | VRNN | P | https://bitbucket.org/gissemari/disaggregation-vrnn |
| [97] | ★ | 2018 | UK-DALE, REDD | R | X | X | X | X | X | | | | | | | | usn | no | P | CNN-s2sub | on/off & P | https://github.com/ch-shin [1] |
| [98] | ★ | 2018 | IDEAL | R | X | | X | X | X | X | | | | | | shower | usn | no | S | CNN-s2sub | P | - |
| [99] | | 2018 | dataport | R | | | | | | | | X | | | | | sn | no | P | LSTM | P, on/off | - |
| [100] | | 2018 | UK-DALE | R | | X | X | | | | | | | | | | ? | no | P | CNN-biLSTM | on/off | - |
| [101] | | 2018 | proprietary | C | | | | | | | | | | | | | sn | no | P | CNN-s2p | P | - |
| [102] | | 2018 | UK-DALE | R | X | X | X | | X | X | | | X | | X | computer | sn | no | P | CNN | on/off$_m$ | - |
| [33] | | 2018 | UK-DALE | R | X | X | X | X | X | | | | | | | | usn | yes | P, P$_{var}$, TofD | CNN-s2p | P, P$_{class}$ | - |
| [103] | | 2018 | proprietary | R | | X | | X | | | | | X | | | bottle warmer, TV | ? | no | P | dAE | P$_m$ | - |

**Table 2.** *Cont.*

| Ref. | Best | Year | Dataset(s) | Setting | DWE | FGE | MWV | WME | KET | SOC | TDR | HVAC | WDR | HPE | Light | Further Appliances | E.Sce. | Aug. | Input | DNN Elements | Output | Code |
|---|---|---|---|---|---|---|---|---|---|---|---|---|---|---|---|---|---|---|---|---|---|---|
| [104] | | 2018 | UK-DALE | R | X | X | X | X | X | | | | | | | | usn | no | P | CNN-VAE | P | - |
| [105] | | 2018 | UK-DALE | R | | X | | X | | | | | | | | | usn | yes | P | GAN | P | https://github.com/KaibinBao [1] |
| [106] | | 2018 | AMPds2 | R | | | | | | | | | X | | | home office | sn | no | P, $T_{out}$ | LSTM | on/off & P | - |
| [107] | | 2018 | dataport | R | X | X | X | | X | X | | | | X | | ~30 more | sn | no | $P_m$ | dAE-LSTM | P, $P_m$ | https://github.com/nlaptev [1] |
| [108] | | 2018 | UK-DALE | R | X | X | X | | X | | | | | | | | sn | no | P | biLSTM, biGRU | P | - |
| [109] | | 2018 | IMD | I | | | | | | | | | | | | | sn | no | P | CNN-wn | P | - |
| [22] | | 2018 | UK-DALE, AMPds2 | R | X | X | X | X | X | X | | | X | | | | usn | yes | P, Q | CNN-dAE | P | [2] |
| [110] | | 2018 | REDD | R | X | X | | | | | | | | X | | | usn | dn | P | CNN-s2sub | P | - |
| [111] | | 2018 | UK-DALE | R | X | X | X | X | X | | | | | | | | usn | no | P | biLSTM, biGRU, CNN-s2p | P | https://github.com/OdysseasKr/online-nilm |
| [112] | | 2018 | UK-DALE | R | X | X | X | X | X | | | | | | | TV | usn | no | P | CNN-dAE | on/off | - |
| [113] | | 2018 | proprietary | C | | | | | | | X | | | | X | | usn | yes | P | CNN-dAE | P | - |
| [21] | | 2018 | UK-DALE, REDD, AMPds | R | X | X | X | X | X | X | X | | X | | | | usn | yes | P | CNN-dAE | P | - |
| [114] | | 2018 | REDD | R | X | X | X | | | X | | X | X | | | | sn | no | na | CNN | on/off$_m$ | - |
| [115] | | 2017 | proprietary | R | | | | X | | X | | | | | | computer, heater | usn | yes | P | dAE | P | - |
| [32] | | 2017 | UK-DALE, REDD | R | | | | X | | X | | X | | | | dehumidifier, toaster, TV | sn | no | P, $P_{var}$ | LSTM | on/off$_m$ | - |
| [116] | | 2017 | REDD | R | X | X | X | | | X | | X | X | | | | sn | no | na | CNN | on/off$_m$ | - |
| [117] | | 2017 | proprietary | R | | X | | | | | | | | | | | usn | no | P, Q, S | CNN-dAE | P & Q | - |
| [118] | ⋆ | 2016 | REDD | R | X | X | X | | | | | | | | | | sn | yes | P | CNN, RCNN, biLSTM, biGRU | $P_{class}$ | - |
| [119] | | 2016 | UK-DALE, REDD | R | X | X | X | X | X | | | | | | | | usn | no | P | CNN-s2s, CNN-s2p | P | https://github.com/MingjunZhong/NeuralNetNilm |
| [120] | | 2016 | UK-DALE | R | | | | | | | | | | | | | sn | no | P | RNN, GRU | on/off | - |
| [121] | | 2016 | UK-DALE | R | X | X | X | X | X | | | | | | | | usn | no | P | CNN-dAE, CNN-LSTM | P | - |
| [122] | | 2016 | REDD | R | X | X | X | | | | | | | | | kitchen outlet | sn | dn | P | HMM-DNN | P | - |
| [123] | | 2016 | dataport | R | | | X | | | X | | | X | | | | usn | no | $P_{2D}$ | CNN | P | - |
| [15] | | 2015 | REDD | R | X | X | X | | | | | | | | | | usn | dn | P | biLSTM | P | - |
| [14] | | 2015 | UK-DALE | R | X | X | X | X | X | | | | | | | | usn | yes | P | CNN-dAE, CNN-biLSTM, CNN-FF | P | https://github.com/JackKelly/neuralnilm |

[1] GitHub page of first author; [2] Experimental framework available upon request.

As the DNN-NILM literature reviewed contains only three publications using data from non-domestic settings (two commercial, one industrial), this distribution means that our review concentrates mainly on domestic NILM.

### 1.3. Methodology

Publications in the scope of this work have been collected in the following ways: Firstly, by systematically checking conference proceedings from the bi-annually 'International NILM Workshop' 2020 to 2016 (nilmworkshop.org, accessed on 11 January 2021) and from the 'ACM International Conference on Systems for Energy-Efficient Buildings, Cities, and Transportation (BuildSys)' 2020 to 2015 (buildsys.acm.org, accessed on 11 January 2021). The first conference is specifically dedicated to NILM, and the second in 2020 featured a dedicated NILM track. Secondly, by searching on Google Scholar and IEEE Xplore® for keyword combinations of 'DNN', 'deep learning', 'NILM', 'non-intrusive', 'load monitoring', and 'load disaggregation'. This search has been done on several occasions and by different persons. Thirdly, we checked very thoroughly all the references in identified papers for anything not yet on our list. While this approach might have missed a few recent publications, we are fairly sure that the survey is quite complete for the past years because of the systematic checking of references. The last iteration of our search process has been performed at the end of November 2020. We resulted with the DNN-NILM publications listed in Table 2, which reflects accordingly the body of work this survey paper is based on. The literature review, discussion, and all conclusions are deduced solely from these publications.

## 2. Fundamentals

As we assume that the reader is knowledgeable about both NILM and Deep Neural Networks, the following sections only skim the corresponding subjects and the reader is referred to relevant literature.

### 2.1. The Disaggregation Problem

The aggregate active power $x_t^a$ of a set of appliances measured at time $t$ can be formally defined as:

$$x_t^a = \sum_{m=1}^{M} y_t^m + \underbrace{\sum_{k=1}^{K} w_t^k + \epsilon_t}_{=e_t} \qquad (1)$$

where $y_t^m$ are the contributions of individual appliances $m$ that have been metered at the time of data acquisition, and $M$ is their total number. The sum over $k$ corresponds to the contribution of $K$ further appliances $w_t^k$ not sub-metered during the measurement campaign. $\epsilon_t$ is a noise term originating from the measurement equipment. In the literature, the NILM problem is typically stated such that the noise term $e_t$ includes the sum over non measured equipment. We explicitly separate the two contributions, as their nature is quite different. We can assume that the measurement noise $\epsilon_t$ is well behaved, i.e., it follows approximately a standard distribution and is small compared to the actual signal. On the contrary, no such assumption can be made about the term $\sum w_t^k$. The contribution from non sub-metered appliances $w_t^k$ typically amounts to a major part of $x_t^a$ and the power distribution is non-Gaussian. From the point of view of disaggregation, the sum over $m$ denotes the appliances that are disaggregated, and the sum over $k$ consists of all the remaining appliances in the aggregate signal. If only a single appliance $y_t^m$ is disaggregated, then $M = 1$.

One goal of energy disaggregation is to determine the individual $y_t^m$ only based on the measurement of the aggregate signal. If machine learning or in particular deep learning is used to solve the problem, this leads to a so-called regression problem. While many authors work with the active power component $x_t^a$ only, other information from the aggregate signal such as, e.g., apparent power, reactive power, or the current can also be used to solve the disaggregation challenge. In the particular case of countries where the residential power

supply is fed on three phases, features from the aggregate power can even be available on all of these three phases.

A second, slightly less challenging goal of energy disaggregation is to find the on/off state $s_t^m$ of appliance $m$ at time $t$ from the aggregate signal. If machine learning is used, this leads to a so-called binary classification problem. In this problem formulation, only the state of the machine will be output. After recognizing the on and off states, the run-time of an appliance can be calculated. By multiplying the run-time by the average energy consumption of a machine, one can still obtain an energy estimation. Such an estimate will be more in line with use cases that require only the average consumption over a certain time period.

### 2.2. Deep Neural Networks

Deep neural networks are a vast subject, and the focus of this review is merely their application to NILM. In this text, we therefore refrain from giving an introduction on DNNs and refer the reader to the following books:

- The book [124] (www.deeplearningbook.com accessed on 11 January 2021) is a very comprehensive resource on the topic, covering the basics up to research topics.
- The book [125] has been written by the initial author of Keras [126], a high level deep learning library. Accordingly, the book gives an applied introduction to deep learning. A lot of emphasis is put on the intuition behind concepts, and the text is interwoven with code examples based on the Keras library.

The references mentioned are the books we found useful in our work. The selection is of course a small subset of the many excellent resources available on the topic.

### 3. Literature Review

When applying DNNs to NILM, many options are available: For example, what data to use, what DNN architecture to employ, how to evaluate the results, and so on. An illustration of these 'degrees of freedom' is given in Figure 1. The subsections below roughly follow the grouping done there. The aim of this section is to provide the reader with an overview on what has already been done in the literature in the scope of this review.

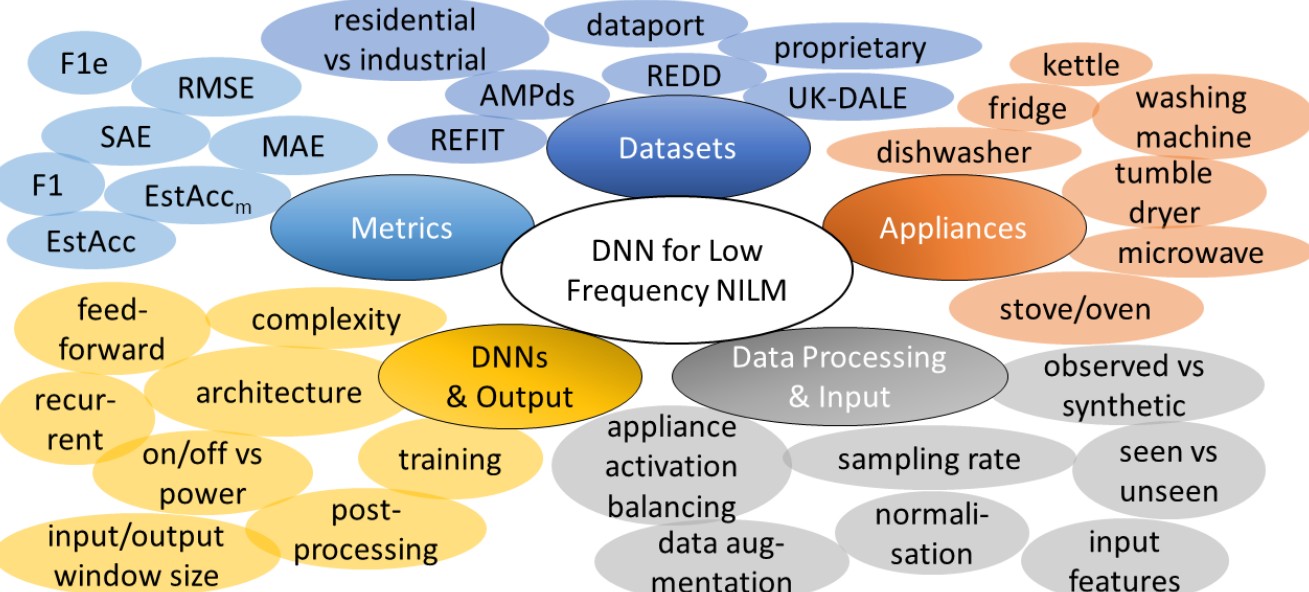

**Figure 1.** Illustration of the main degrees of freedom for DNN-NILM research. Colors indicate a loose grouping and should not be understood as a taxonomy.

### 3.1. Datasets and Appliances

DATASETS: The number of NILM datasets has been increasing over the last years, see [127,128] for recent overviews and [129–131] for the most recent published datasets we are aware of. In Table 3, we characterize only the publicly available datasets that have been used in the reviewed studies. The datasets at the beginning of the table are those more frequently used: Both UK-DALE and REDD were employed in approximately 40 and 30 studies, respectively, followed by AMPds, REFIT, and dataport each employed around 10 times. The ECO dataset is used three times, and other datasets are only used once or twice. The Industrial Machines Dataset (IMD) is to our knowledge the only available open industrial dataset. All remaining open datasets were measured in a residential setting. There are also a number of studies based on proprietary datasets measured in different settings: Nine residential, two commercial, one industrial, and one with dc-appliances. While not explicitly the scope of this review, the distribution of the employed datasets means that our review concerns mostly domestic NILM. Table 2 lists the datasets employed by each reviewed publication.

APPLIANCES: Appliances that have been disaggregated in the corresponding publications are listed in Table 2. The most investigated residential appliances in decreasing order are: dishwasher, fridge, microwave, washing machine, kettle, stove/oven/cooker, tumble dryer, HVAC, washer-dryer, heat pump, and light. Further electrical loads that appear fewer than ten times in the literature are given in the column 'Further Appliances' of Table 2. A few publications concentrated either on commercial or industrial applications using mostly proprietary datasets. These publications are marked in the column 'Setting' of Table 2.

**Table 3.** Main characteristics of the open datasets used in the reviewed DNN-NILM literature, see Table 2. Datasets closer to the top have been employed in more studies. *Type* indicates the type of the dataset: $R \rightarrow$ residential, $R_s \rightarrow$ synthetic residential, $I \rightarrow$ industrial. IMD is, to our knowledge, the only publicly available industrial dataset. '#H' and '#A' mean number of houses and appliances, respectively. '*Agg*' and '*Appl*' stand for 'aggregate' and 'appliance', respectively. For the IDEAL dataset, available information has been extracted from [98]. The authors plan to release the dataset.

| Name | Country Code | Year | Type | #H | #A | Summed up Duration [d] | Agg Sampling | Appl Sampling |
|---|---|---|---|---|---|---|---|---|
| UK-DALE [132] | GBR | 2017 | R | 5 | 109 | 2247 | 6 s, 16 kHz | 6 s |
| REDD [133] | USA | 2011 | R | 6 | 92 | 119 | 1 Hz, 15 kHz | $\frac{1}{3}$ Hz |
| AMPds(2) [134–137] | CAN | 2016 | R | 1 | 20 | 730 | 1 min | 1 min |
| REFIT [138] | GBR | 2016 | R | 20 | 177 | 14,600 | 8 s | 8 s |
| dataport [139] | USA | 2015 | R | 1200+ | 8598 | 1,376,120 | 1 Hz, 1 min | 1 Hz, 1 min |
| ECO [140] | CHE | 2016 | R | 6 | 45 | 1227 | 1 Hz | 1 Hz |
| DRED [141] | NLD | 2014 | R | 1 | 12 | 183 | 1 Hz | 1 Hz |
| Enertalk [127] | KOR | 2019 | R | 22 | 75 | 1714 | 15 Hz | 15 Hz |
| HES [142] | GBR | 2010 | R | 251 | 5860 | 15,976 | 2–10 min | 2–10 min |
| IDEAL | GBR | - | R | - | - | - | 1 Hz | 1 or 5 Hz |
| IMD [143] | BRA | 2020 | I | 1 | 8 | 111 | 1 Hz | 1 Hz |
| PLAID [144] | USA | 2014 | R | 65 | 1876 | 1–20 s | - | 30 kHz |
| SynD [130] | AUT | 2020 | $R_s$ | 1 | 21 | 180 | 5 z | 5 Hz |

### 3.2. Data Processing

Raw datasets can be employed differently for training and evaluating DNN-NILM approaches. Below, we review different aspects.

#### 3.2.1. Training and Evaluation Scenarios

Training and evaluation of NILM algorithms can be done under different scenarios. Typical scenarios appearing in the literature are defined in the following.

OBSERVED VS. SYNTHETIC: In a *synthetic* scenario, the term $\sum w_t^k$ in Equation (1) is set to zero.

Corresponding data are typically created by summing up the power consumption from individual appliances. Only the measurement noise $\epsilon_t$ is therefore included in the noise term $e_t$. In an *observed* scenario, the noise term $e_t$ also includes further appliances that have not been measured individually, i.e., $\sum w_t^k \neq 0$. The *synthetic* scenario can be

considered a laboratory setting for a basic assessment of algorithms. Data in a real scenario will typically be *observed*.

We use here the terms *observed* and *synthetic* scenario equivalently to *noised* and *denoised* scenario, as these scenarios are commonly referred to in the literature. We introduce this new nomenclature because we believe that (i) the original terms are rather misleading for readers with less experience in the field, and (ii) the proposed terms express the essential difference between the two scenarios much more precisely.

SEEN VS. UNSEEN VS. CROSS-DOMAIN TRANSFER: The terms seen and unseen are used in the context of the evaluation of NILM algorithms. In the *seen* case, an algorithm is evaluated on new data from households that it has already been trained on. The resulting score gives, therefore, an indication on how well the trained algorithm can detect a particular appliance. *Unseen* means, that the algorithm is evaluated on data from a new household that was not available in the training data. This scenario tests the capability of algorithms to detect an appliance type [145]. Corresponding test results indicate the performance of a pre-trained model that is deployed on data from houses previously not seen during training. For the *cross-domain transfer learning* [91] scenario, the unseen house is taken from a different dataset. This scenario tests the transferability of the tested approach to an even more diverse setting as in the unseen case: Data could have been metered by different electrical meters or could originate from a different country. To our best knowledge, this scenario has only been investigated in [39,42,63,71,91]. The different scenarios are illustrated in Figure 2. The column 'Evaluation Scenario' in Table 2 lists the scenarios employed for the reviewed references.

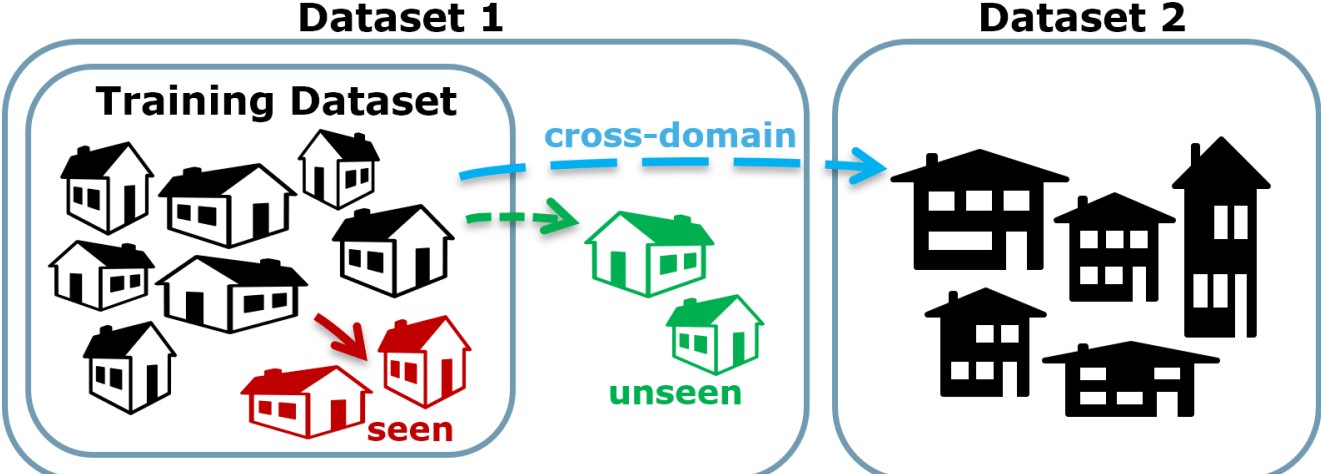

**Figure 2.** Different NILM evaluation scenarios: *seen*: the algorithm is evaluated on new data from a house that was already available during training; *unseen*: the algorithm is evaluated on data from a house not seen during training; *cross-domain transfer learning*: the algorithm is evaluated on data from a different dataset.

### 3.2.2. Preprocessing

Before data can be used by the DNNs, the raw data are transformed. Below we discuss typical data transformation steps employed in the literature.

RESAMPLING, FORWARD-FILLING, AND CLIPPING: The sampling frequencies of the published datasets are given in Table 3. As datasets exhibit missing values due to measurement or transmission equipment failures and a jitter in the timestamps, resampling is used to obtain evenly sampled data. While the range of sampling frequencies in the reviewed literature extends from $\frac{1}{3600}$ Hz [92,107] to 10 Hz [75], the large majority of the reviewed works employ either $\frac{1}{60}$ Hz or values between 1 and $\frac{1}{10}$ Hz. It is noteworthy that in two cases, data were upsampled to have a higher frequency than the original dataset [36,112]. Results on the influence of the sampling frequency on disaggregation results are presented in different studies [51,58,75,77]. Most of these studies find a marked dependence on

the device [51,58,75]. This can be attributed to certain devices exhibiting more frequent fluctuations that get lost at lower resolution. Ref. [75] analyzes sampling rates from 10 Hz down to 0.03 Hz for on/off classification and energy estimation for TV, washing machine, rice cooker. They find that to prevent performance loss for the classification and regression tasks, the sampling rates should be at least 1 Hz and 3 Hz, respectively. Ref. [58] compares results obtained with 10 s and 1 min sampling intervals. The authors find "that the performance for dishwashers remains comparable while the performance for washing machine and washer dryer deteriorates dramatically". The publication [51] focuses exclusively on the influence of the sampling rate on the performance. The authors conclude that data sampled at 1/30 Hz might be sufficient to run NILM at high accuracy. It is important to note, that [51] did, contrary to [58,75], fix the number of inputs to the DNNs instead of the temporal window. Consequently, the temporal window seen by the network differs in this study depending on the sampling rate. Finally, [77] investigates the influence of the sampling rate in case of appliance on-event detection.

Short spans of missing data attributed to WiFi connectivity problems are forward-filled by many authors with the last available measurement. Typically, up to three minutes of missing data are filled in this manner [14]. In case of measurements exceeding the rating of the employed meter, values are clipped.

NORMALIZATION In the DNN-NILM literature, the input normalization for the DNNs comes in two main flavors:

$$x_{stdScaled} = \frac{x - \bar{x}}{\sigma(x)} \tag{2}$$

$$x_{minmaxScaled} = \frac{x - x_{min}}{x_{max} - x_{min}} \tag{3}$$

where $x$ and $x_{Scaled}$ are the input windows (see Section 3.3.1) before and after normalization. $\bar{x}$ corresponds to a mean value over the input. Different strategies have been employed: Most approaches calculate the mean over the complete training set so that the training data are centered. Other strategies center the data per house (see, e.g., [75]) or per input window (see, e.g., [14,107]). $\sigma(x)$ denotes the standard deviation, which is typically calculated on the complete training set. Alternatively, each input window was divided by the standard deviation from a random subset of the training data [14]. $x_{max}$ and $x_{min}$ correspond to maximal and minimal values. These values can be maximal or minimal values of the training dataset, parameters fixed by the authors [53], or quantile values [40]. In order to make the statistics of the data less sensitive to outliers, [44] transformed them with an *arcsinh* before normalizing. Some authors also normalized the target values for the training of the DNNs. While some publications mention that different normalization strategies were tried out, only two studies report on the influence of normalization strategies on training efficiency and testing performance: [34] finds that instance normalization [146] performs better than batch normalization [40,147] concludes that L$_2$-normalization works best.

### 3.2.3. Activation Balancing

In NILM literature, the time interval between an appliance being switched on and off is referred to as an *activation*. Domestic appliances exhibit typically one, up to several activations per day. Usually, the run-time of appliances is low compared to the time they are switched off. For the training of machine learning algorithms, one is consequently faced with a skewed dataset that contains only a few samples of the running appliance. To compensate, several authors balance samples with and without a (partial) activation during training [14,22,34,35,39,58,60,63,75,89,105,110,112,119,123]. The majority of works nevertheless train the models using the available data, without taking care of the class imbalance. In the scope of this review, we are only aware of [34], which investigates the effect of the ratio between samples with and without an activation on training results. They found that in case of batch normalization [147], the accuracy strongly decreased at a ratio

of one to five, whereas for instance normalization [146], the performance increased slightly up to the largest tested ratio of one to seven. In general, it remains unclear how exactly activation balancing influences the disaggregation quality and model convergence speed.

### 3.2.4. Data Augmentation

A common strategy in deep learning to overcome few labeled data samples or under-represented classes employs data augmentation. It describes the process of transforming existing measured data or creating synthetic new data in order to achieve DNNs that generalize better. Recent overviews for data augmentation in the domains of computer vision and time series can be found in [148–150].

In the reviewed DNN-NILM literature, we see different data augmentation variants: First, some approaches train in a synthetic scenario, see Section 3.2.1. Only synthetic data consisting of the summed up loads from appliance sub-meters are used to train and test the algorithms. Such publications are denoted with a '*dn*' in the column 'data augmentation' of Table 2. A second group of publications train on measured aggregate data but add synthetic data—also created by summing up sub-metered load curves—to increase the size of the training set. Four authors added individual activations from appliances to a measured aggregate [6,58,66,115]. Finally, some authors employed specialized strategies: The authors of [35] found that by adding varying offsets specifically to the on state of the fridge, they were able to greatly enhance the corresponding disaggregation performance. So-called 'background filtering' has been proposed by [69] to remove all windows in the aggregate load curve that contain the target appliance. Activations from the target appliance are then added randomly to the filtered aggregate to create synthetic data for training. The authors of [44] use data obtained from SMACH [151], a tool that generates synthetic data based on time of use surveys and real appliances signatures. They compare scenarios with different amounts synthetic data and find good generalization performance for models trained only on synthetic data. We are not aware of any study that compares different data augmentation strategies.

### 3.3. Input

### 3.3.1. Shape

The vast majority of approaches take as a continuous regularly sampled window from the time series of the aggregate measurement data input for the DNNs. The range of employed window lengths extends from 90 s [75] to around 9 h [94,98] or even 24 h [66,107]. It is important to note that the number of input samples to the neural networks, i.e., the number of neurons in the first layer, depends on the sampling rate. Extreme values for the size of the input layer are 5 [100], 7 [83], and 10800 [112]. The influence of the window length on the disaggregation performance at a fixed sampling rate is investigated in [52,62,97]. While the investigations have been done on different datasets and sampling rates ([97] → UK-DALE, REDD at 3 and 6 Hz, [62] → REFIT at 10 Hz, and [52] → DRED at 1 Hz), all authors find that the optimal window length depends on the appliance.

Few authors transform the time series data into a two-dimensional representation before feeding it into a DNN. Corresponding publications are marked with '$P_{2D}$' in the column 'Input' of Table 2: [38,71,123] use the Grammian Angular Field (GAF) [152] to transform a continuous part of the aggregate measurement into a two-dimensional representation which is then fed to a convolutional neural network (CNN). Ref. [38] additionally compares the performance of the GAF with the Markov Transition Field [152] and a Recurrence Plot [153] and finds that the GAF outperforms the other imaging techniques in the vast majority of experiments. Ref. [92], on the other hand, arranged hourly consumption readings into two dimensions by setting the hour of the day as the x-coordinate and the day as the y-coordinate. The authors found that the optimal size of the first filter amounts to $7 \times 7$, allowing the filter to learn weekly correlations.

A minority of works use a DNN to classify events extracted by a previous detection stage [77,80,114,116] or to classify the on–off status directly from a single time step [82]. These publications are marked with 'na' in the column 'Input' of Table 2.

### 3.3.2. Features

The active power from the aggregate measurement is the only input for most of the reviewed works. There are, however, a number of papers that extended the input to further features: Reactive and apparent power, current, first-order difference of the active power signal, power factor, the variant power signature, and different time-based features have been used additionally. A noteworthy case is the input of the aggregate power from multiple neighboring buildings which, according to [107], lead to considerable performance improvements. Input features of the reviewed publications are marked in the column 'Input' of Table 2. See also Section 4.2 for a discussion of the benefits of multiple input features.

### 3.4. Deep Neural Networks

### 3.4.1. Architectures Elements

In Table 2, column 'DNN Elements', we summarize proposed DNN architectures based on a set of DNN building blocks or elements. Naturally, this attempt can only be a coarse approximation of the encountered diversity. It still provides a high-level view on what has been tried out. We mention only original architectures proposed by the authors. Models from earlier authors or baselines are not listed. As a consequence the column is empty in several cases, e.g., where the authors compare previous works. Looking at Table 2, we observe that starting with the year 2018, feedforward elements—in particular convolutional elements—gained in popularity. These elements are used roughly twice as much as recurrent elements. In the same time span, advanced DNN elements such as generative adversarial networks (GAN) and attention were also adapted to NILM.

Below, we give a description of the DNN elements used in Table 2 to describe the proposed models:

- FF → Feedforward network, see, e.g., [154]. We use the abbreviation in case a simple multilayer feedforward network is a major component of the network. It is not used for dAE, CNNs, or LSTM networks that contain, e.g., only a final feedforward layer for classification.
- dAE → Denoising autoencoder [155]. We use the abbreviation for architectures made up of encoder-decoder architectures. In the context of NILM, denoising AE are used to separate the appliance's signal from the rest of the aggregate signal.
- VAE → Variational autoencoder. Special variant of the AE that encodes the latent variables as distributions [156,157].
- CNN → Convolutional neural network, see, e.g., [154]. We use the abbreviation for networks that employ CNN layers. s2p, s2sub, s2s, and wn are more detailed subclasses.
- s2s → The abbreviation is used for networks that map from an input sequence to an output sequence with identical length. It is only used if the output of the network is active power or the on/off state of an appliance.
- s2sub → The abbreviation is used for networks that map from an input sequence to an output sub-sequence, i.e., the output length is smaller that the input length. It is only used if the output of the network is active power or the on/off state of an appliance.
- s2p → The abbreviation is used for networks that map from an input sequence to a single output value. It is only used if the output of the network is active power or the on/off state of an appliance.
- wn → Wavenet [158] inspired architectures. In particular, dilated convolutions and gating mechanisms are important elements of these architectures.
- att → This abbreviation subsumes different variants of attention mechanisms. Specifically, ref. [55] used the mechanism defined in [159], ref. [35] employed attention from [160], and [72] from [161].

- RNN → The abbreviation is used for networks that employ vanilla recurrent neural networks, see, e.g., [154].
- (bi)LSTM → The abbreviation is used for networks that employ (bidirectional) long short-term memory cells [162,163].
- (bi)GRU → (bidirectional) gated recurrent unit [164].
- GAN → The abbreviation is used for networks that employ elements from generative adversarial networks [165].
- RCNN → recurrent CNN [166].
- HMM-DNN → combination of Hidden Markov Model with a classic feed-forward network [122].
- VRNN → recurrent variational neural network [167].

In case different elements have been combined, they are joined with a hyphen '-'. That means, e.g., CNN-dAE corresponds to a dAE that includes convolutional layers.

### 3.4.2. Training and Loss Functions

DNN gradient descent has its own set of hyperparameters such as the type of optimizer, number of training epochs, or early stopping criterion. As these elements are not specific to the NILM problem, we just mention a few specific points in this section. There have been several authors that optimized (training) parameters of the proposed networks by automatic means. For example, grid search [72,92], Hill Climbing [41], and Bayesian optimization [42,75,90,102] have been employed. [118] investigated different variants of curriculum learning [168]. In this type of learning, samples are not randomly presented to the DNN, but organized in a meaningful order, the intuition being that humans learn by mastering concepts with increasing difficulty. Contrary to this intuition, ref. [118] finds that easy samples hinder training, and the author used synthetic training data composed from sets of more than 7 appliance sub-meters.

A key element of DNN optimization is the loss function that guides the optimization process. The vast majority of works employ either the mean absolute error (MAE) or the mean squared error (MSE) in case of power disaggregation and the cross entropy loss for on/off classification. Recent works also investigate alternative loss functions: Quantile regression [169] was employed by [46,59]. The authors of [59] found that their proposed loss increased the performance of two state-of-the-art models compared to the MSE loss. Some works [34,35,39,43] employ GAN loss functions—called 'adversarial loss' in [35]—that classify if the output of the regression DNN is a real or fake appliance load curve. This loss should make outputs more realistic and help especially in case of datasets of limited size. Finally, ref. [55] introduced a loss composed from four different terms: In addition to the MSE, a Kullback–Leibler divergence loss, a soft-margin loss, and the MAE are used. To our knowledge, a systematic comparison of loss functions for DNN-NILM approaches has not been published.

### 3.5. Output

With respect to their output, we observe four different dimensions along which DNN-NILM approaches can be distinguished: A first dimension is the number of time steps that are disaggregated by the DNN models in a single go, be it a sequence, subsequence, or a single value. This information is available through the abbreviations *s2s*, *s2sub*, and *s2p* in the column 'DNN Elements' of Table 2, see also Section 3.4.1. The second dimension concerns the type of inferred output. With the exception of [70], where location data (*location*) are combined with the aggregated power consumption, and [95], where a DNN infers state changes in the aggregate power (*stateChange*), the goal of DNN-NILM approaches is either to infer the on/off state or the energy usage of an appliance. We mark this information in column 'Output' of Table 2 with the abbreviations *on/off* and *P*, respectively. Naturally, this dimension is mostly coupled with the third distinction, whether the learning problem is formulated as a classification or regression task. However, there are four works [33,61,68,118] where power values are clustered into groups, and the power

regression problem is recast into a classification task. These references are marked with $P_{class}$. Lastly, we can distinguish between approaches that learn on a single task and those learning on multiple tasks simultaneously, i.e., perform multi-task learning. The majority of approaches train one model for each appliance to be disaggregated. A sizable number of approaches infer, however, the on/off state or power disaggregation of multiple appliances simultaneously. These cases are marked with a subscript $m$ in the 'Output' column of Table 2. Where multi-task learning is done on different modes, the corresponding outputs are jointed with an '&' in Table 2: [35,63,97,106] trained networks on both on/off and active power data. Ref. [117] used both active and reactive power of an appliance as target, and [59] used three targets, i.e., the aggregate power ($P_{total}$), the appliance power ($P_{app}$), and the difference between the two ($P_{rest}$). Finally, ref. [46] took multi-task learning furthest by simultaneously learning on both on/off states and active power of multiple appliances.

### 3.5.1. Post-Processing

Different variants to process the output from the DNN-NILM approaches have been proposed in the literature. In cases where the DNN is of type *s2s* or *s2sub* and disaggregation is done by moving the input window a single time step at a time, the network will deliver n predictions for each time step, where n is the length of the output window. In order to obtain a single prediction, many authors used the mean, e.g., [14,22,35,89,119] or the median, e.g., [21,72]. In [21], the authors find that networks underestimate the power of appliances if activations are only partially in the input window. As a consequence, the mean also underestimates the ground truth and [21] proposes to use the median instead, which is less impacted by this problem. As the authors of [39] use a GAN, they only use the disaggregated signal for which the discriminator outputs the highest probability of being a true sample.

Some authors note that the models produce noisy output, e.g., in the form of sporadic activations either too short or too frequent for the target appliance. Ref. [37] filters out such events with the same approach as for activation detection in the ground truth data. Similarly, [36] removes all activations of an appliance that are shorter than those found in the ground truth data. Depending on the metric, the reported improvement ranges from 28% to 54%. Refs. [58,89] go one step further and train a second DNN to suppress spurious activations. According to [58], the additional DNN leads to "significant performance boosts".

### 3.6. Evaluation Metrics

The performance of NILM algorithm is assessed in various ways. The interested reader is referred to [18,145]: Ref. [18] provides a comprehensive review and discussion of employed metrics, and ref. [145] proposes a set of metrics to assess the generalization ability of NILM aglorithms. In the following, we only repeat the definition of the mean absolute error (MAE) and the $F_1$-score. In the reviewed literature, these were the most encountered metrics to assess the estimated energy consumption and on/off status of an appliance, and we use them for our comparison in Section 4.1.

$$\text{MAE} = \sum_t^T \frac{|y_t - \hat{y}_t|}{T} \tag{4}$$

where the sum goes over $T$ time steps, and $y_t$, $\hat{y}_t$ correspond to the measured and estimated power consumption, respectively. In this publication, we use Watts as the unit for the MAE.

$$F_1 = 2 \cdot \frac{P \cdot R}{P + R} \tag{5}$$

where precision $P = TP/(TP + FP)$ and recall $R = TP/(TP + FN)$ with $TP$, $FP$, and $FN$ denoting true positives, false positives, and false negatives, respectively [18].

## 4. Discussion and Current Research Gaps

The following sections discuss different aspects of the reviewed literature. Each section concludes with a paragraph on current research gaps we see concerning the discussed topic.

### 4.1. Performance Comparison

One of the basic questions that accompanied us throughout this literature review was: "What is the most promising approach or classes of approaches?" As the last section hints, there is no straightforward answer to that. Too many degrees of freedom (see Figure 1) make the approaches differ in so many ways that a comparison based solely on the results presented in the publications can only give indications. For that purpose, the MAE and $F_1$ score were extracted from the reviewed publications wherever available. (The data is available on our GitHub account. The link is provided in the 'Supplementary Materials'.) These two metrics are the most applied performance measures in case of energy estimation and on/off state classification, respectively. Figures 3 and 4 each display the best reported results split up by dataset and appliance. Only results from the *observed*, *unseen* evaluation scenario are given. This scenario was selected as it is closest to an actual application of DNN-NILM algorithms, see Section 3.2.1. The graphs only include results from approaches proposed in the corresponding publications: Results from baselines, or approaches from earlier work that were used for comparison, are not included. Appliances with a single result in the displayed range are excluded. We observe that the results for kettle and microwave are overall better and not as distributed as those from the other displayed appliances. We believe this is because of their simpler nature: Both kettle and microwave are appliances with only two states whereas dishwasher, washing machine, and fridge (to a certain degree) have a more diverse load signature.

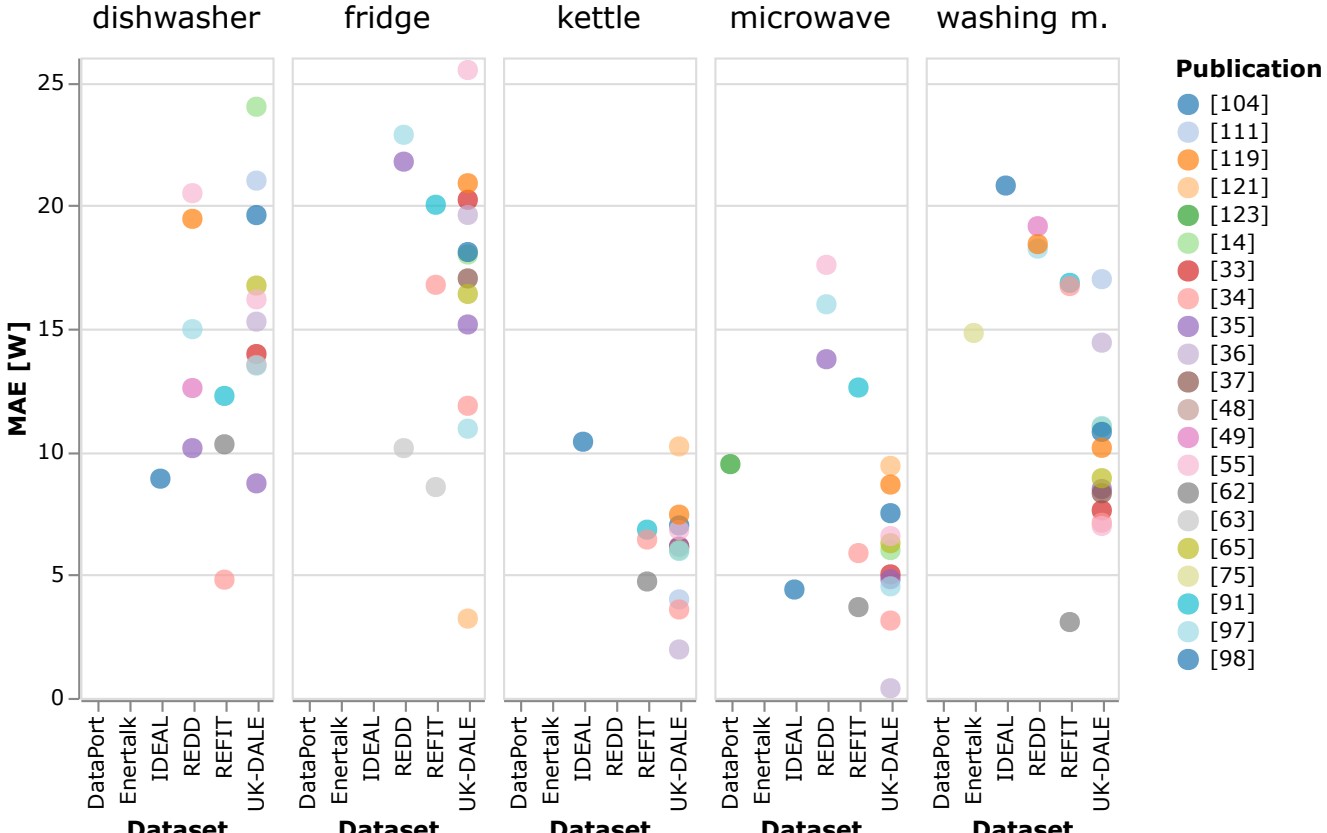

**Figure 3.** Minimal reported MAE for the corresponding dataset and appliance. Only results from the *observed*, *unseen* evaluation scenario have been included. Only approaches proposed by the authors in the corresponding publications are taken into consideration (i.e., no baselines or models from the state-of-the-art). Results have been split according to the appliance and employed dataset. Please note that appliances with a single result in the selected range are not shown.

We caution the reader *not to interpret the displayed values as the result of a direct comparison under identical conditions*. Results have been generated under broadly differing settings, see Table 2. One key difference is that evaluation data varied strongly between publications. While results in Figures 3 and 4 are not directly comparable, we try to identify common elements of successful approaches. For that purpose, we sorted the results for each appliance (irrespective of the dataset) and took the top quarter of the results. Depending on the appliance, a quarter consisted of four to six results in case of the MAE and two to four in case of the $F_1$-score. We then evaluated the number of times a publication appears in these results. Those with *more than one count* are [34,35] (five times), [62] (four times), and [36,63,97,98] (two times) in case of MAE, and [63] (six times), [118] (three times), and [36,37,64] (two times) for the $F_1$-score. These publications have been marked in column 'Best' of Table 2.

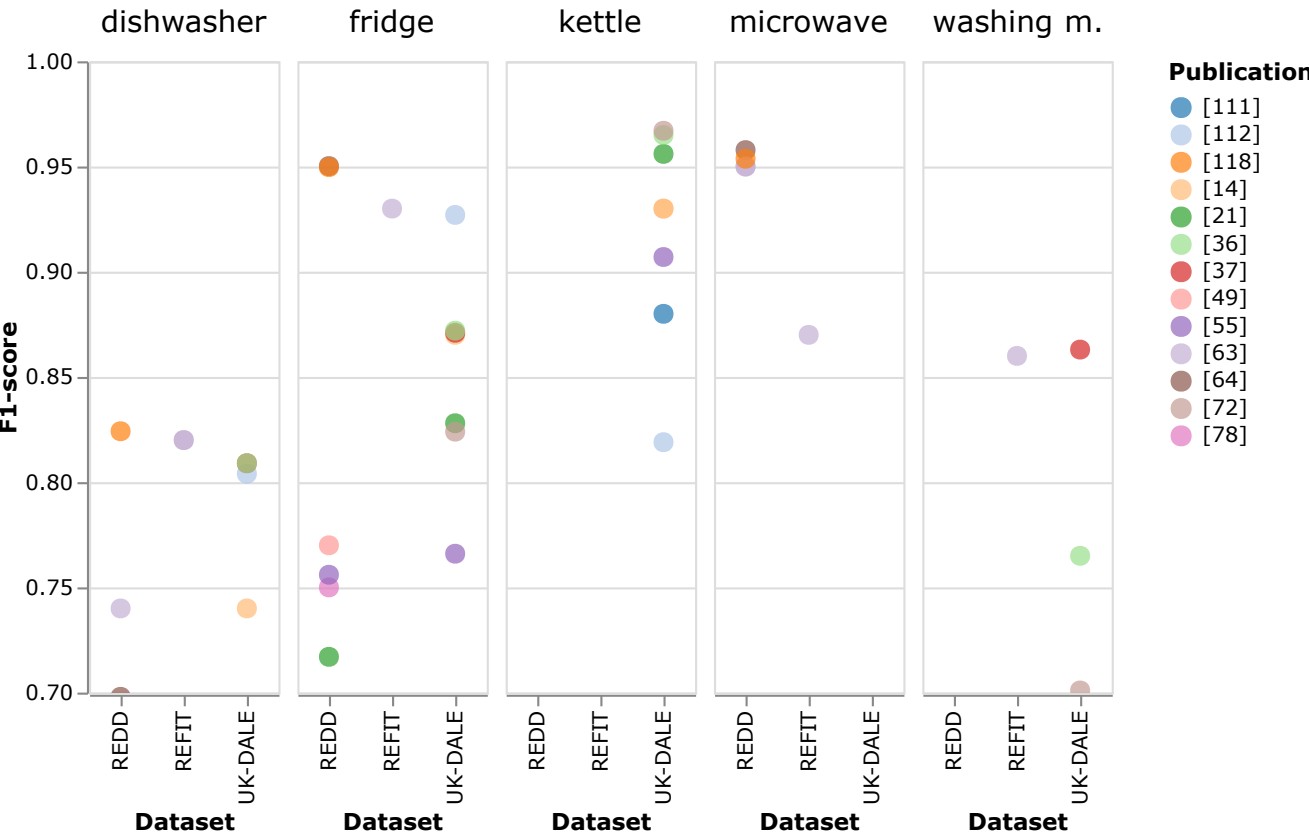

**Figure 4.** Maximal reported $F_1$-score for the corresponding dataset and appliance. Only results from the *observed*, *unseen* evaluation scenario have been included. Only approaches proposed by the authors in the corresponding publications are taken into consideration (i.e., no baselines or models from the state-of-the-art). Results have been split according to the appliance and employed dataset. Please note that appliances with a single result in the selected range are not shown.

Based on these publications, we make the following observations:

- With the exception of [37], who used 60 s intervals, the best results are based on data with sampling intervals up to 10 s.
- The architectures of [34,35,37,62,98] contain all elements that allow an output neuron to have a large field of view. Refs. [35,62,98] use dilated convolutions that allow for an exponential increase in the field of view of deeper network layers, see, e.g., WaveNet [158] that served as template in case of [62]. The models from [35,37,62] all concatenate and process at some stage in the network output from multiple layers that each have widely varying fields of view. This is called 'scale-awareness' by [35] and 'temporal pooling' by [37]. Ref. [34] adopts the U-net [170], an encoder–decoder architecture originally proposed for image segmentation.

- The approaches of [34,35] employ a GAN setup. This means that the network does not (only) learn based on a static loss function, but it rather exploits the 'feedback' of a second network that classifies if the output of the first one is real or fake.
- Four approaches [35,37,63,97] use multi-task learning for network training: ref. [37] trains the model to classify the on/off state of multiple appliances at the same time, whereas [35,63,97] simultaneously train their networks to estimate the energy usage and the on/off state of single appliances. In the latter three cases, this is done with networks that have one branch for each objective. The final result is then obtained by combining the branches.
- Refs. [36,37] use very similar post-processing schemes that remove sporadic activations, see Section 3.5.1. Ref. [36] reports improvements in the range of 28% to 54% compared to results without post-processing.

We see the following limitations in the previously performed comparison of the MAE and $F_1$-scores: As already pointed out, results have been obtained through wildly different procedures. It is also questionable if these metrics are the most relevant, they have simply been chosen as the ones mostly provided in the reviewed literature. That means also that publications reporting other metrics do not appear in the evaluation. With these words of caution said, we still believe the previous observations provide some value.

With respect to the initial question "How does the performance of approaches compare?" and the performed literature review, we identify several challenges worth addressing by the research community: We observe that the experiments performed in the reviewed literature are not always well specified with respect to the 'degrees of freedom' mentioned in Section 3. We hope future works profit from our listing of available options and specify their decisions clearly. Besides, we see several additional steps at different levels that could lead to a better comparability:

- Beside the metrics motivated by the intended use case, published approaches should report a set of standard metrics. Simply based on their availability in the reviewed literature, we propose to use the MAE and $F_1$-score in case of energy estimation and on/off state classification tasks, respectively.
- We see a great potential in defining a standard evaluation protocol that defines training and testing folds for cross-validation of models per dataset. Of course it should respect particularities of the NILM setting such as the evaluation scenarios, and it would ideally be in a machine readable form such as proposed for the ExperimentAPI of NILMTK [23].
- Authors would ideally publish the code for the employed models and experiments as already done by several authors, see Table 2. Based on that code, retraining for comparison with new approaches is greatly simplified.
- Beside the model specification as code, trained models could also be released as is done in the vision community, see, e.g., [171]. We are only aware of the trained models of [49] that have been publicly released.
- As has already been noted by other authors [19,140], a comparison of NILM approaches on the same terms is very beneficial for NILM practitioners. In case of DNN-NILM approaches, we feel that this work can serve as a good basis for such an undertaking providing a comprehensive overview on the relevant literature and published code. A Python framework for comparison has already been published in [23] and applied to the analysis of four DNN-NILM (beside classical) approaches in [24]. A possible route to stimulate such comparisons would also be to organize challenges like the one which took place in 2020 [172]. In the computer vision domain, for example, the ImageNet challenge was a great driver for innovation.

### 4.2. Multiple Input Features

While the reactive power has already been used in the founding works of NILM [1,2], most authors take the active power as the only input for disaggregation, see Section 3.3.2.

Therefore, we raise the question: "Can we find evidence that multiple input features benefit DNN-NILM performance?"

As can be seen in the overview Table 2, different authors employed alternative input features. The following authors report results from a comparison of input features [22,36,89,94,106,107,117]. Ref. [117] was the first DNN-NILM approach we are aware of that used multiple input features. Unfortunately, these results do not allow separation of the influence of input features from multi-task learning because the two are always used in conjunction. Ref. [22] explicitly exploits reactive power (Q). The authors evaluate the impact of Q on the $F_1$-score within the AMPds and UK-DALE datasets. Over the investigated appliances, they find an average improvement of around 12.5% in the seen and 8% in the unseen evaluation scenario. (Average $F_1$-scores for P in seen and unseen scenarios are 0.68 and 0.58, respectively.) Interestingly, the reported improvement is small or negative for purely resistive loads, such as kettle and electric oven. We hypothesize that in such cases, the reactive power provides no information, but purely noise. Refs. [89,94,106] all worked with the AMPds dataset. This data set contains measurements from a single house, thus all results stem from seen evaluation scenarios. Ref. [94] compares two feature sets with the help of the estimation accuracy: For the combination P and Q versus P alone, the authors find an improvement of 6%. For the combination P, Q, current (I), and apparent power (S), the improvement is slightly higher at 7%. (The estimation accuracy in case of P alone amounts to 0.83.) Ref. [89] investigates the same feature set P, Q, I, S versus P based on three different performance measures, namely the MAE, root mean square error (RMSE), and the normalized RMSE. In this work, the improvements with the additional features are much larger: For all measures, the average improvement is around 40% to 50%. (The MAE, RMSE, and normalized RMSE averaged over all investigated appliances and models for P alone amount to 36.7 W, 122.8 W, and 0.75, respectively.) The temperature as a supplementary feature has been used by [106]. The authors find that the disaggregation of 'heat pump' and 'home office' works 3% and 4% better based on the $F_1$-score and estimation accuracy, respectively. (The $F_1$-score and estimation accuracy averaged over the two appliances in case of P alone amount to 0.87 and 0.91, respectively.) The authors of [107] find that providing the aggregate electrical consumption from neighbors as additional features leads to performance improvements of 17% and 31% with and without multi-task learning, respectively. (The symmetric mean absolute percentage error (SMAPE) for P alone amounts to 23% and 38% in the respective cases.) Finally, ref. [36] calculates the mutual information between P, Q, S, I, voltage, the power factor of the aggregate measurement, and P of the appliance as a feature selection step. Voltage is the least informative feature, and is therefore dropped for the subsequent evaluations.

Previously mentioned improvements have been calculated from the original values reported by the authors according to the following formula

$$Improvement = \frac{Perf_{addFeatures} - Perf_{onlyP}}{Perf_{onlyP}}. \tag{6}$$

where $Perf_{onlyP}$ and $Perf_{addFeatures}$ correspond to the performance (measured in any measure) of the approach based only on P and additional features, respectively. In cases where smaller values indicate better performance of a measure (e.g., MAE), we swapped $Perf_{addFeatures}$ and $Perf_{onlyP}$ to always result with a positive value for improved results.

Based on the results presented, we conclude that features beside P can improve disaggregation performance. No conclusions about the amount of improvement can, however, be made, as the spread of the results is quite broad. For the time being, we can only speculate about possible reasons. It might be a worthwhile investigation to examine what kind of factors, e.g., architectures, can make the most out of the information from features beside P. With the exception of those in [22], all results originate from seen evaluation scenarios. That effectively means that additional features help to estimate the power usage of a particular appliance. However, it is unclear how much they help to disaggregate an appliance *type*, see Section 3.2.1. Non DNN-NILM approaches already

employed a very broad feature set [13]. Compared to this breadth, DNN-NILM approaches tested a very limited set of options. It would be interesting to see a systematic investigation on a broader feature set.

### 4.3. Multi-Task Learning

If a machine learning model trains on separate but related tasks, this process is referred to as multi-task learning. For a good introduction and overview on the topic with respect to deep learning, the reader is referred to [173]. The NILM problem is suitable to be framed as a multi-task learning problem: The column 'Output' in Table 2 lists the different variants that have been employed in the reviewed literature. We asked ourselves: "Can we find evidence that multi-task learning leads to superior performance compared to single task learning in the case of the DNN-NILM approaches?"

Based on the literature review, we found the following: Ref. [91] trains a CNN on the washing machine, freezes the parameters of the convolutional layers, and retrains afterwards only the final, fully connected layer for other appliances. The authors find that the results of this approach are comparable to standard training. This finding suggests that the learned features of different appliances are similar and can be shared between appliances. Simultaneous learning on different appliances could therefore make features more robust and lower the requirements on the amount of training data. A large improvement from joint learning on multiple appliances is also reported by [107] and, as was already mentioned in Section 4.1, four of the best approaches [35,37,63,97] use multi-task learning for network training. Only the authors of [49] report a general decrease in performance of multi-task learning models with respect to their single-task counterparts. They propose to employ a different architecture or share less layers between appliances as a remedy. Due to the presented observations and the general benefits of multi-task learning presented in [173], we conclude that multi-task learning is beneficial for DNN-NILM approaches. As has also been noted by [49], we see the additional benefit of multi-task learning in a reduced computational burden for edge devices because a major amount of computations for disaggregation can be shared between several applications.

### 4.4. Parameter Studies

As visualized in Figure 1, there are many degrees of freedom for DNN-NILM approaches. In Section 3, we listed the many options for the corresponding aspects that have already been tried out. Looking at the literature, however, we see a lack of understanding of the influence of the available options. Therefore, we want to stress the need and value of parameter studies for future research activities in the DNN-NILM field.

For example, in case of the data *sampling rate* and *window length*, several authors looked at the influence of these two parameters on the models performance, see Sections 3.2.2 and 3.3.1. There exists, however, no study that jointly investigates these two tightly connected parameters (maybe even on different datasets and based on different models). Similarly, we see potential in a systematic comparison between different normalization (Section 3.2.2), activation balancing (Section 3.2.3), data augmentation (Section 3.2.4), and post-processing (Section 3.5.1) strategies, as well as loss functions (Section 3.4.2).

### 4.5. Applied DNN-NILM

The best results of current DNN-NILM approaches are very promising, see Section 4.1. However, there are different aspects of relevance for an actual deployment of DNN-NILM approaches that have not yet been well investigated in the literature. In the following subsections, we motivate corresponding aspects and subsequently point out connected research gaps.

### 4.5.1. Data Scarcity

For a practical application of NILM, we see one of the main challenges in the scarcity of labeled data. While this challenge is not specific to DNN approaches, we think that recent developments in semi-supervised deep learning might be adaptable to NILM and could then be a great opportunity to tackle the problem of data scarcity for practical applications. In the following, we will first detail the challenge and subsequently formulate possible future research directions.

Net2Grid is a company providing NILM services to utilities. In a presentation, they stressed the point that "accurate NILM requires [...] a lot of high-quality data" [174]. Specifically, the company bases their NILM service on data from *hundreds* of houses. They also emphasized the point that machines with different programs or settings exhibit very variable load patterns and therefore need many observed cycles. The authors of [75] investigate how the disaggregation error of a DNN-NILM approach depends on the number of distinct houses used for its training. In agreement with Net2Grid, they find that for the washing machine, an appliance with a variety of programs, the disaggregation error steadily decreases with each house added to the training dataset without any sign of saturation until 40 houses, which was the maximum used for the investigation. Thus, both sources indicate that complex machines require a large variability in the training data to successfully generalize on unseen data. This observation is at the core of what we call 'data scarcity'.

These findings indicate that a company that wants to start a NILM service first has to obtain data from hundreds of houses. A possible source are public NILM datasets including aggregate and appliance consumption, see, e.g., [18,127] for an overview. While for some simple appliances these public datasets are certainly sufficient, the data are too restricted if we want to disaggregate appliances with variable load patterns [75,174]. Furthermore, appliances such as heat pumps or charging stations for electric vehicles are almost absent from public datasets. The only alternative is to engage in a measurement campaign involving *hundreds* of houses. This is an expensive and time consuming undertaking, as the metering, appliance-specific sub-metering, and the corresponding infrastructure has to be installed and maintained. It is also worth noting that, even if large datasets covering a large variety of appliances would be recorded, new devices come to the market continuously. This means that the effort for data collection is actually a recurrent one.

As DNNs are particularly data hungry, the problem of a shortage of labeled data has recently obtained a lot of attention by the research community in the computer vision and natural language processing (NLP) domain. A promising remedy to the situation is semi-supervised deep learning [175]. We see a lot of potential transferring these developments to the NILM domain as unlabeled data—the data obtained from the smart meter—are relatively easy to access compared to sub-metered ground truth data.

In the reviewed DNN-NILM literature, semi-supervised deep learning techniques have so far been employed by the following authors: Ref. [103] used the ladder network [176]. The presented results give no clear indication that unsupervised data actually improved the performance. These results might be caused by the relatively simple DNN. The authors of [66] trained an autoencoder on unlabeled data to subsequently use the learned embedding in a supervised training setting. This work was done on aggregate data with 15 min resolution that naturally led to large estimation errors. Ref. [81] derived their classification approach from the mean teacher approach [177], while ref. [67] adopted virtual adversarial training [178]. Both works present evidence that the disaggregation performance of the semi-supervised approaches improves compared to a strictly supervised settings. However, experiments were only conducted on data from houses already seen during training, and no conclusion can be drawn about improved generalization on previously unseen houses.

There is an increasing field of newer semi-supervised DNN approaches from the vision domain ready to be adapted to the NILM problem [175]. A particularly successful [179] semi-supervised strain of research is called *consistency learning* [175]. The method's main assumption is that a small perturbation or realistic transformation applied to a data point should not have an influence on the prediction. DNNs are then trained to provide a consistent

output for an unlabeled data point and its perturbed version. Most recent publications demonstrate that for image classification it is feasible to get close to the performance of a supervised approach with one order of magnitude fewer labeled samples [180,181]. While some of the consistency learning approaches seem to be well adaptable to NILM, there remain many open questions—to name a few: What type of consistency loss should be used in case of NILM? What types of data augmentation strategies should be employed? The last question is of particular interest because [180] demonstrated that the 'quality' of data transformations is the key for significant performance gains. While data augmentation was used in various DNN-NILM approaches, see Table 2, we are not aware of any work that did a detailed investigation of this aspect.

In summary, we see in the application of semi-supervised DNN approaches many worthwhile research questions and a great opportunity to tackle the problem of data scarcity for practical NILM applications.

### 4.5.2. NILM on Embedded Systems

If one imagines a NILM deployment at scale, the amount of data to transfer, store, and process becomes an important factor. In the reviewed DNN-NILM literature, different aspects of such a deployment have been addressed. The work of [93] investigated data reduction policies: Different sampling strategies for data compression ($\frac{1}{4}$ to $\frac{1}{20}$ compression) in combination with DNN inference are tested. The authors found that the best sub-sampling policies outperform results with original sampling rates. Another option is to process the data directly in or close to the electric meter and only relay disaggregated high level information. That means that the DNN-NILM inference has to work on an embedded system, even though that can be quite challenging in terms of computational, storage, and energy resources. This direction has been investigated by [47–49,65,106]: Ref. [106] is to our knowledge the first to publish the implementation of DNN-NILM inference on an embedded device. Both [106] and later [65] used for that purpose a Raspberry Pi computer. Ref. [47] uses an efficient MobileNet [182,183] inspired DNN for disaggregation and compresses it by lowering the precision from32 to bit floating point (used for training) to 8-bit integer representation by means of the TensorFlow Lite library. The resulting model was then evaluated with the Android SDK. The authors report that "disaggregation accuracy deviates up to ≈9.4% from original disaggregation model, but, on average, remains satisfactory". Both refs. [48] and [49] investigate different pruning methods based on the network from [119]. Pruning methods aim at reducing neurons in the network that contribute little to the final output. The final goal is to result with sparse networks that have lower storage and computational requirements, but similar performance compared to the original networks. Both publications found that networks can be heavily pruned with only a slight decrease in performance: Ref. [48] reports a reduction of the number of network weights by 87% and [49] reports a 100-fold reduction in model size and a 25-fold reduction in inference times. Ref. [49] additionally investigates multi-task learning and vector decomposition as further paths towards efficient computations in embedded systems.

While the DNN-NILM community has taken first steps towards an implementation on embedded devices, the corresponding research field for DNNs in the vision and speech domain is vast, see, e.g., [184]. There remain therefore a multitude of research questions in this direction. From our perspective, an interesting question would be to see how best performing approaches (see Section 4.1) could be adapted to embedded devices because the architectures of these approaches are more elaborate than the ones used by [47–49,106].

### 4.5.3. 3-Phase Data

In some European countries, such as, e.g., Switzerland, residential power supply arrives in three phases at the master distribution board (breaker panel) and is then split into single phases. As a consequence, measurements from the electrical metering infrastructure are in principle also available on three phases. With respect to a practical NILM application, this additional information makes the problem at first glance easier to solve, as there are

on average one third as many devices connected to each phase compared to households attached to a single phase. However, the challenge comes in the form of multi-phase appliances such as heat pumps, pool pumps, electrical heat storage radiators, or charging stations for electrical vehicles. These appliances require NILM algorithm to combine information from all three phases. When considering an approach that should perform on any households, the main challenge is that multi-phase devices can be connected in arbitrary permutations. Thus, the result of the DNN-NILM approach needs to be invariant to these permutations.

We are not aware of any DNN-NILM publication that works on 3-phase data and tackles the raised challenge (This might partially be because there are currently only few datasets with 3-phase information. We are aware of iAWE [185], ECO [140], and BLOND [186]). The desired permutation invariance is analog to the required rotational invariance in computer vision: An object needs to be recognized as such independently of its orientation in the image. This analogy points also to possible future research questions: Could permutation invariance be obtained by training a DNN with augmented data? Could the symmetry be directly anchored in the layer of the neural network via Group Equivariant Convolutions, see, e.g., [187,188]? These are convolutional layers specially designed to produce the same result for data subject to a group of symmetry operations. How do these two solution approaches compare to each other with respect to performance and complexity?

## 5. Outlook

Looking into the future, we can imagine different scenarios and directions for the (DNN-)NILM field. With the rapid development of the Internet of Things, we can well think of future appliances which are aware of their own current (and possibly future) energy consumption and feature a communication interface to relay this information to the outside world. In this scenario of energy-aware appliances, NILM would become obsolete. As this scenario would require a business case for appliance manufacturers and standards for interfaces and protocols, chances are good that this state will not be reached in the near future. We believe that the rapid increase of computing power in edge devices will have a much more immediate impact. Edge nodes will soon be able to perform DNN-NILM close to the meter without the need to transfer data to a cloud computing service. The culmination of this trend would be complex NILM algorithms that run directly on meter hardware, maybe even on the raw high frequency measurement data. Developing this scenario even further, one could imagine that NILM algorithms learn and improve on local data. For this to work, the learning problem will first have to be formulated in a way that the data available on the meter can be used for further improvements. A standard supervised training approach does not seem to be feasible. Furthermore, local improvements of the model will ideally also be made available to other smart meters. This concept of local learning with global exchange of improvements is a nascent research field called Federated Learning, see [189,190].

## 6. Conclusions

Summarizing, this publication presents a review on the DNN-NILM literature. The scope of this review comprises publications that employ deep neural networks to disaggregate appliances from low frequency data, i.e., data with sampling rates lower than the AC base frequency. Our motivation for the scope is our conviction that plenty of applications could benefit from NILM, coupled with the observation that low frequency data will most likely be available at scale in the near future and the enormous success of DNNs in other application domains. We systematically discuss the many degrees of freedom of these approaches and what has already been tested and tried out in the literature along these dimensions. One of the main contributions is Table 2, which gives a structured overview of the main characteristics of all reviewed DNN-NILM approaches. The review part is followed by a discussion of selected DNN-NILM aspects and corresponding research gaps.

We present a performance comparisons with respect to reported MAE and $F_1$-scores and observed different recurring elements in the best performing approaches, namely data sampling intervals below 10 s, a large field of view, the usage of GAN losses, multi-task learning, and post-processing. Subsequently, the benefit of multiple input features and multi-task learning and related research gaps has been discussed, the need for comparative studies has been highlighted, and the missing elements for a successful deployment of DNN-NILM approaches have been pointed out. Finally, we also outline potential future scenarios for the NILM field. This contribution is currently missing in the literature, and can therefore be of value. We conclude that there remain many worthwhile research questions to be pursued.

**Supplementary Materials:** To facilitate future work based on the data collected for this publication, we release Table 2 as a MS Excel file. We also provide data and code that was used to generate Figures 3 and 4. All data and code is available at https://github.com/ihomelab/dnn4nilm_overview accessed on 11 January 2021.

**Author Contributions:** Conceptualization, P.H., A.R. and A.P.; data curation, P.H.; formal analysis, P.H.; funding acquisition, A.R. and A.P.;investigation, P.H.;project administration, A.R.;supervision, A.R. and A.P.;visualization, P.H.;writing, P.H.;writing—review and editing, P.H., A.C., A.R. and A.P. All authors have read and agreed to the published version of the manuscript

**Funding:** This research was funded by Innosuisse—Schweizerische Agentur für Innovationsförderung, grant number 36152.1 IP-EE and the Lucerne University of Applied Sciences and Arts. The APC was funded by the Lucerne University of Applied Sciences and Arts.

**Institutional Review Board Statement:** Not applicable.

**Informed Consent Statement:** Not applicable.

**Data Availability Statement:** The data presented in this study are available at https://github.com/ihomelab/dnn4nilm_overview accessed on 11 January 2021.

**Acknowledgments:** We want to express our gratitude to Gianni Gugolz, who supported us compiling Table 3.

**Conflicts of Interest:** The authors declare no conflict of interest.

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
