# Peer review of "Review on Deep Neural Networks Applied to Low-Frequency NILM"

_energies, doi:10.3390/en14092390_

Round 1
Reviewer 1 Report
The reviewer agrees with the authors, that the NILM field had rapidly increased in term of publications number, and it is a suitable time for performing a review study, however the article has still big room for improvement:
- The authors define low frequency data as of sampling rates lower than the AC base frequency. The reviewer is aware of a convention in NILM community that low-frequency data is less than 1 Hz. The author might need to refer to that somewhere in the paper.
- The abbreviation must be defined when mentioned the first time, please spell it (e.g. GAN mentioned in the abstract). It is preferable in the abstract not to include abbreviated terms, it is not mandatory it just looks elegant.
- Please re-write your abstract, try to more particularly define the overall purpose of the study, the basic design of the review, more importantly the major findings or trends found as a result of your review. Remember, paper abstract should be a self-contained section!
- Try to read your own paper again and omit the non-scientific looking phrases. For example, in line 26, would change "let us clarify" by "it is essential to clarify...", this is not mandatory but adds scientific spirit to language.
- Line 39 "similar appliances", similar in what sense? Same type? similar activation shapes?
- Paragraph (line 58 - 76) is ended weakly. This paragraph should be the driver for the reader to continue reading the paper. Try to emphasize why the review worth reading, how it is particularly different from other reviews. May be it is a good time to make a review for NILM work, because of the recent updates in the field, emphasize something.
- The methodology part needs to be reviewed, the reviewer agrees that the international NILM workshop can contain a lot of NILM related works, however there is a huge portion of other NILM works are not presented there. IEEE Journals, other IEEE conferences, IEEE explore is full of that. Those repositories and publications contain also high-quality works. This why Table 1 looks not very sensible for the reviewer, one can list papers close to those numbers just from the memory! The author might reconsider the review methodology. As two simple examples:
- A good comparative study for denoising autoencoders is missing: https://www.sciencedirect.com/science/article/abs/pii/S0378778817314457 “Bonfigli, Roberto, et al. "Denoising autoencoders for non-intrusive load monitoring: improvements and comparative evaluation." Energy and Buildings 158 (2018): 1461-1474.”
- The authors mentioned to the best of their knowledge that only 3 papers are investigating cross-domain transfer scenario. In fact some other papers studied that. However, a recent paper analyzing this transferability is here: https://ieeexplore.ieee.org/document/9302933. They also used data augmentation. It is also missed in the discussion about GANs and data augmentation. “A. M. A. Ahmed, Y. Zhang and F. Eliassen, "Generative Adversarial Networks and Transfer Learning for Non-Intrusive Load Monitoring in Smart Grids," 2020 IEEE International Conference on Communications, Control, and Computing Technologies for Smart Grids (SmartGridComm), Tempe, AZ, USA, 2020, pp. 1-7, doi: 10.1109/SmartGridComm47815.2020.9302933.”
The reviewer recommends to cross check with other repositories.
- It is preferred to use the term Deep Learning instead of Deep Neural Networks. As DL is a broader field of ML.
- line 169, the author might just refer to the on-line version in the references part.
- Please rephrase paragraph line 166. It is not clear.
- Taxonomy in fig. 1 needs to be reviewed! not clear looks less professional.
- For the industrial dataset, author can also check also: de Mello Martins, Pedro Bandeira, Raphael Guimaraes Duarte Pinto, and Pedro Bittencourt e Silva. "Load Disaggregation of Industrial Machinery Power Consumption Monitoring Using Factorial Hidden Markov Models."
- Table 2, you might add a column telling if the dataset is household or industrial.
- paragraph 202 is very short.
- The Evaluation Metrics subsection should be revised, other metrics are recently started to be used in the literature, e.g. SAE in "D’Incecco, Michele, Stefano Squartini, and Mingjun Zhong. "Transfer learning for non-intrusive load monitoring." IEEE Transactions on Smart Grid 11.2 (2019): 1419-1429.
- In table 3, reference 133 used their own dataset, try to check and fill the missing entry in the table.
- An extensive discussion section is there, however it has to be revised and increase its readability. This is the author’s most original contribution. Might also map the discussion to the broad machine learning schemes, e.g. unsupervised and supervised.
- The review misses the authors’ reflections about what do they think about the potential of using deep learning methods for on-line NILM?
- Under the embedded systems restrictions, how do the authors think that deep learning models are restricted by the model size?
- What are the generalizability risks of DL?
Author Response
Dear Reviewer
Dear Editorial Team
We thank the reviewer for their effort and feedback to improve the quality of
the submitted manuscript. Below each comment is cited and followed by our
response.
1. The authors define low frequency data as of sampling rates lower than the
AC base frequency. The reviewer is aware of a convention in NILM community
that low-frequency data is less than 1 Hz. The author might need to refer
to that somewhere in the paper.
Response to Comment 1:
We added a footnote explaining the motivation for our slightly different
definition of low- and high-frequency data.
2. The abbreviation must be defined when mentioned the first time, please
spell it (e.g. GAN mentioned in the abstract). It is preferable in the
abstract not to include abbreviated terms, it is not mandatory it just
looks elegant.
Response to Comment 2:
We followed the reviewer's suggestion and spelled out abbreviations in the
abstract.
3. Please re-write your abstract, try to more particularly define the overall
purpose of the study, the basic design of the review, more importantly the
major findings or trends found as a result of your review. Remember, paper
abstract should be a self-contained section!
Response to Comment 3:
We extended the abstract with a sentence stating the overall purpose.
4. Try to read your own paper again and omit the non-scientific looking
phrases. For example, in line 26, would change "let us clarify" by "it is
essential to clarify...", this is not mandatory but adds scientific spirit
to language.
Response to Comment 4:
The suggested change was done.
5. Line 39 "similar appliances", similar in what sense? Same type? similar
activation shapes?
Response to Comment 5:
The term "similar appliances" has been removed and replaced with "appliances of
the same type".
6. Paragraph (line 58 - 76) is ended weakly. This paragraph should be the
driver for the reader to continue reading the paper. Try to emphasize why
the review worth reading, how it is particularly different from other
reviews. May be it is a good time to make a review for NILM work, because
of the recent updates in the field, emphasize something.
Response to Comment 6:
The text was extended to more clearly state why the review is worth reading.
The section "Contriutions" follows now directly on the paragraph with previous
review papers.
7. The methodology part needs to be reviewed, the reviewer agrees that the
international NILM workshop can contain a lot of NILM related works,
however there is a huge portion of other NILM works are not presented
there. IEEE Journals, other IEEE conferences, IEEE explore is full of that.
Those repositories and publications contain also high-quality works. This
why Table 1 looks not very sensible for the reviewer, one can list papers
close to those numbers just from the memory! The author might reconsider
the review methodology. As two simple examples:
A good comparative study for denoising autoencoders is missing:
https://www.sciencedirect.com/science/article/abs/pii/S0378778817314457
“Bonfigli, Roberto, et al. "Denoising autoencoders for non-intrusive load
monitoring: improvements and comparative evaluation." Energy and Buildings
158 (2018): 1461-1474.” The authors mentioned to the best of their
knowledge that only 3 papers are investigating cross-domain transfer
scenario. In fact some other papers studied that. However, a recent paper
analyzing this transferability is here:
https://ieeexplore.ieee.org/document/9302933. They also used data
augmentation. It is also missed in the discussion about GANs and data
augmentation. “A. M. A. Ahmed, Y. Zhang and F. Eliassen, "Generative
Adversarial Networks and Transfer Learning for Non-Intrusive Load
Monitoring in Smart Grids," 2020 IEEE International Conference on
Communications, Control, and Computing Technologies for Smart Grids
(SmartGridComm), Tempe, AZ, USA, 2020, pp. 1-7, doi:
10.1109/SmartGridComm47815.2020.9302933.”
The reviewer recommends to cross check with other repositories.
Response to Comment 7:
We agree with reviewer 1, that there is a huge portion of NILM work outside of
the mentioned conferences and were under the impression that google scholar
would consistently turn this up. Upon the reviewers feedback, we double checked
with IEEE Xplore and found indeed additional relevant papers: The review was
therefore extended with 13 additional publications and the text
- wherever appropriate - extended correspondingly. With respect to the specific
references pointed out by the reviewer:
* "Bonfigli, Roberto, et al. "Denoising autoencoders for non-intrusive load
monitoring: improvements and comparative evaluation." Energy and Buildings
158 (2018): 1461-1474.”
This publication has already been included in the submitted manuscript as
reference number 20.
* “A. M. A. Ahmed, Y. Zhang and F. Eliassen, "Generative Adversarial Networks
and Transfer Learning for Non-Intrusive Load Monitoring in Smart Grids," 2020
IEEE International Conference on Communications, Control, and Computing
Technologies for Smart Grids (SmartGridComm), Tempe, AZ, USA, 2020, pp. 1-7,
doi: 10.1109/SmartGridComm47815.2020.9302933.”
This publication is now included in the review. We thank the reviewer for
pointing it out.
8. It is preferred to use the term Deep Learning instead of Deep Neural
Networks. As DL is a broader field of ML.
Response to Comment 8:
We respectfully disagree with this comment, since the works that we referenced
all used structures which can be described as deep neural networks, as opposed to the classical 'shallow' ones.
9. line 169, the author might just refer to the on-line version in the
references part.
Response to Comment 9:
Thanks for the good suggestion. We added the link to the online version after
the reference.
10. Please rephrase paragraph line 166. It is not clear.
Response to Comment 10:
The paragraph has been adapted and better structured.
11. Taxonomy in fig. 1 needs to be reviewed! not clear looks less professional.
Response to Comment 11:
We mention in the caption of figure 1, that is is *not* a taxonomy. We have
adapted the caption to make this clearer. Additionally, we changed the naming of different elements to make them more consistent with names of the sections in the literature review.
12. For the industrial dataset, author can also check also: de Mello Martins,
Pedro Bandeira, Raphael Guimaraes Duarte Pinto, and Pedro Bittencourt e
Silva. "Load Disaggregation of Industrial Machinery Power Consumption
Monitoring Using Factorial Hidden Markov Models."
Response to Comment 12:
After double-checking, we come to the conclusion that this conference paper refers to the dataset that we call "IMD". Our reference points to the data published in a repository.
13. paragraph 202 is very short.
Response to Comment 13:
We acknowledge the comment, but feel that the paragraph serves its purpose.
14. Table 2, you might add a column telling if the dataset is household or
industrial.
Response to Comment 14:
We followed the reviewer's suggestion and added a column 'Type' indicating if a
dataset is of residential or industrial origin.
15. The Evaluation Metrics subsection should be revised, other metrics are
recently started to be used in the literature, e.g. SAE in "D’Incecco,
Michele, Stefano Squartini, and Mingjun Zhong. "Transfer learning for
non-intrusive load monitoring." IEEE Transactions on Smart Grid 11.2
(2019): 1419-1429.
Response to Comment 15:
We are aware that many more evaluation metrics are available and we refer to
corresponding reviews in the first sentences of this section. We only repeat
the definition of those that will be used in section "4.1 Performance
Comparison". This information has been added in the text. These are - as
already indicated - the ones most consistently reported in most of the reviewed
publications.
16. In table 3, reference 133 used their own dataset, try to check and fill the
missing entry in the table.
Response to Comment 16:
The entry has been filled.
17. An extensive discussion section is there, however it has to be revised and
increase its readability. This is the author’s most original contribution.
Might also map the discussion to the broad machine learning schemes, e.g.
unsupervised and supervised.
Response to Comment 17:
The first part of the comment is rather general. After reviewing our text, we
decided to do larger adaptions in section "4.5.1 Data Scarcity". As our fourth
author - Andrew Paice - is native English speaker and for him the text was
otherwise well understandable, no further action was taken based on this
comment. The second part of the comment is an interesting suggestion, but such a discussion would enlarge the scope too much and we decided not to include such a discussion.
18. The review misses the authors’ reflections about what do they think about
the potential of using deep learning methods for on-line NILM?
Response to Comment 18:
We agree with the reviewer that this is an interesting question. We decided not
to include such a discussion as it would further extend the already lengthy
document.
19. Under the embedded systems restrictions, how do the authors think that deep learning models are restricted by the model size?
Response to Comment 19:
This depends of course very much on the embedded hardware which can have largely varying specifications. We refrain therefore from discussing this point in the review paper and point the reviewer to the referenced literature where specific hardware has been used.
20. What are the generalizability risks of DL?
Response to Comment 20:
In the section "4.5.1 Data Scarcity", we detail the main problem we see with
respect to generalizability of DNN-NILM approaches.
We are convinced that the inclusion of new papers in the review and the other changes have further increased the quality of the manuscript
and hope that these changes are satisfactory also for the editorial board.
Yours sincerely,
Patrick Huber
Reviewer 2 Report
The authors did a great work in this paper. They have managed to detect most of the literature in the field and their analysis is up-to-date and helpful.
1. There are other review papers in the NILM area, so I believe that the authors need to emphasize on the "contributions" section what is the main difference and contribution of their work against them.
2. Although the literature survey is extended, there are a few recent papers that need to be mentioned:
a) Azizi, Elnaz, Mohammad TH Beheshti, and Sadegh Bolouki. "Event Matching Classification Method for Non-Intrusive Load Monitoring." Sustainability 13.2 (2021): 693.
b) Athanasiadis, Christos, et al. "A Scalable Real-Time Non-Intrusive Load Monitoring System for the Estimation of Household Appliance Power Consumption." Energies 14.3 (2021): 767.
c) Piccialli, Veronica, and Antonio M. Sudoso. "Improving Non-Intrusive Load Disaggregation through an Attention-Based Deep Neural Network." Energies 14.4 (2021): 847.
3. The amount of references you managed to collect and analyse here is great. Please make sure that all the references are properly mentioned. There are a few that actually refer to workshop presentations that cannot be retrieved by a link or something?
Author Response
Dear reviewer
Dear editorial team
We thank the reviewer for their effort and feedback to improve the quality of
the submitted manuscript. Below each comment is cited and followed by our
response.
1. There are other review papers in the NILM area, so I believe that the
authors need to emphasize on the "contributions" section what is the main
difference and contribution of their work against them.
Response to Comment 1:
Section "1.1 Contribution" has been moved directly after the description of
previous review papers and the corresponding gaps have been stated more
clearly.
2. Although the literature survey is extended, there are a few recent papers
that need to be mentioned:
a) Azizi, Elnaz, Mohammad TH Beheshti, and Sadegh
Bolouki. "Event Matching Classification Method for Non-Intrusive Load
Monitoring." Sustainability 13.2 (2021): 693.
b) Athanasiadis, Christos, et al. "A Scalable Real-Time Non-Intrusive Load
Monitoring System for the Estimation of Household Appliance Power
Consumption." Energies 14.3 (2021): 767.
c) Piccialli, Veronica, and Antonio M. Sudoso. "Improving Non-Intrusive
Load Disaggregation through an Attention-Based Deep Neural Network."
Energies 14.4 (2021): 847.
Response to Comment 2:
Thank you for pointing us out these interesting recent papers. All references
have been published after the cutoff date on the end of November 2020, see
section "1.3 Methodology". We refrain from changing this because it would
certainly mean to include even more recent papers.
3. The amount of references you managed to collect and analyse here is great.
Please make sure that all the references are properly mentioned. There are a
few that actually refer to workshop presentations that cannot be retrieved
by a link or something?
Response to Comment 3:
We updated references wherever possible with doi numbers. For those without, we checked that they are easily retrievable with google or google scholar.
We are convinced that our changes have further increased the quality of the manuscript and hope that these changes are satisfactory also for the editorial board.
Yours sincerely,
Patrick Huber
Reviewer 3 Report
Dear Authors,
The paper provides a very good literature review on important nowadays topic as deep neural networks applied to low-frequency non-intrusive load monitoring.
Current research work provides a deep study of facilities used for that application.
The authors highlight in a comprehensive way the major research gaps concerning the application of deep neural networks for non-intrusive load monitoring.
The strength part of the paper is examples of the possible role of multiple input features and multi-task learning on non-intrusive load monitoring performance.
The conclusions answer the aims of the study and supported by references.
The manuscript is well organized, and it is easy to read. References are relevant and referenced correctly.
Author Response
Dear reviewer
Dear editorial team
We thank the reviewer for their effort of reviewing the paper and feedback.
Based on the feedback from reviewer 3, we did not feel that any changes were
necessary.
We are convinced that our other changes have further increased the quality of the manuscript and hope that these changes are satisfactory also for the editorial board.
Yours sincerely,
Patrick Huber
Round 2
Reviewer 1 Report
Please review the paper for the purpose of the readability in mind. You might spot some places where the reader might have some difficulty to understand your points.